


# The effects of diachronous surface uplift of the European Alps on regional climate and the oxygen isotopic composition of precipitation

Daniel Boateng[1*], Sebastian G. Mutz[1], Armelle Ballian[2,3], Maud J. M. Meijers[4], Katharina Methner[2,5], Svetlana Botsyun[6], Andreas Mulch[2,3], Todd A. Ehlers[1]

[1]Department of Geosciences, University of Tübingen, Tübingen, Germany
[2]Senckenberg Biodiversity and Climate Research Centre, Frankfurt am Main, Germany
[3]Goethe University Frankfurt Institute of Geoscience, Frankfurt am Main, Germany
[4]Institute of Earth Sciences, NAWI Graz Geocenter, University of Graz, Austria
[5]Department of Geophysics and Geology, University of Leipzig, Leipzig, Germany
[6]Institute of Meteorology, Freie Universität Berlin, Berlin, Germany

*Correspondence to*: daniel.boateng@uni-tuebingen.de

**Abstract.** The European Alps are hypothesized to have experienced diachronous surface uplift in response to post-collisional processes such as, e.g., slab break-off. Therefore, understanding the geodynamic and geomorphic evolution of the Alps requires knowledge of its surface uplift history. This study presents the simulated response of regional climate and oxygen isotopic composition of precipitation ($\delta^{18}O_p$) to different along-strike topographic evolution scenarios. These responses are modeled to determine if diachronous surface uplift in the Western and Eastern Alps would produce $\delta^{18}O_p$ signals in the geologic record that are sufficiently large and distinct for stable isotope paleoaltimetry. This is tested with a series of sensitivity experiments conducted with the water isotope tracking atmospheric General Circulation Model (GCM) ECHAM5-wiso. The topographic scenarios are created from the variation of two free parameters, (1) the elevation of the West-Central Alps and (2) the elevation of the Eastern Alps. Results suggest significant changes in the spatial patterns of $\delta^{18}O_p$, the elevation-dependent rate of change in $\delta^{18}O_p$ ("isotopic lapse rate"), near-surface temperatures, precipitation amounts, and atmospheric circulation patterns in response to the different scenarios. The predictions for the diachronous surface uplift experiments are distinctly different from simulations forced with present-day topography and for simulations where the entire Alps experience synchronous surface uplift. Topographic scenarios with higher elevations in the West-Central Alps produce higher magnitude changes and an expansion of the affected geographical domain surrounding the Alps when compared to present-day topography. Furthermore, differences in $\delta^{18}O_p$ values of up to -2 to -8 ‰ are predicted along the strike of the Alps for the diachronous uplift scenarios, suggesting that the signal can be preserved and measured in geologic archives. Lastly, the results highlight the importance of sampling far-field and low-elevation sites using the δ-δ paleoaltimetry approach to discern between different surface uplift histories.



# 1 Introduction

The topographic evolution of orogens over geological time is controlled by geodynamic processes (e.g., crustal thickening, lower crustal flow, subduction erosion, and lithospheric delamination) and climate-driven surface processes (e.g., erosion, denudation, and other earth surface processes) (e.g., Valla et al., 2021; Whipple, 2009; Ehlers and Poulsen, 2009). Knowledge of past topography contributes to the understanding of climate-tectonic interactions, the tectonic evolution of collisional domains, and the distribution of biodiversity (Allen, 2008; Clark, 2007; Mulch, 2016; Rowley and Garzione, 2007; Antonelli et al., 2018; Mulch et al., 2018). Paleoelevations have been reconstructed for numerous high mountain ranges like the Himalayas and the Tibetan Plateau (e.g., Garzione et al., 2000; Quade et al., 2011; Gébelin et al., 2013; Rowley and Currie, 2006; Ding et al., 2022; Spicer et al., 2021), the North America Cordillera (e.g., Huntington et al., 2010; Chamberlain et al., 2012; Mulch et al., 2006), the Andean Plateau (e.g., Garzione et al., 2008; Barnes and Ehlers, 2009; Garzione et al., 2014; Mulch et al., 2010; Pingel et al., 2016; Sundell et al., 2019) and areally smaller orogens such as the Pyrenees (Huyghe et al., 2012), the Sierra Nevada Mountains of California (e.g., Mulch et al., 2006; Mulch et al., 2008), the Southern Alps of New Zealand (Chamberlain et al., 1999), the Taurides of Turkey (Meijers et al., 2018) and the European Alps (Campani et al., 2012; Fauquette et al., 2015; Krsnik et al., 2021). Past surface elevations have been inferred using a variety of methods, such as foliar physiognomy (Forest et al., 1999), stomatal density in fossil leaves (McElwain, 2004), vesicularity of basaltic flows (Sahagian and Maus, 1994), and water isotopologues from lacustrine and pedogenic carbonates and authigenic minerals (e.g., Kohn and Dettman, 2007; Quade et al., 2007; Rowley and Garzione, 2007; Mulch and Chamberlain, 2007). Among these techniques, stable isotope paleoaltimetry is the most widely used due to the robust systematic inverse relationships between elevation and oxygen ($\delta^{18}$O) and hydrogen ($\delta$D) isotopic composition of meteoric waters reflected in geologic archives such as, e.g., paleosol carbonates. This $\delta^{18}$O-elevation relationship (or isotopic lapse rate) is commonly attributed to the preferential rainout of heavy isotopologues of water from air masses ascending over topography and is described physically as Rayleigh distillation (Gat, 1996). However, numerous non-linear climatic processes, such as surface recycling, aridity, vapor mixing, variability in moisture source, and precipitation dynamics, can also influence the isotopic lapse rate and thus complicate stable isotope paleoaltimetry estimates (Ehlers and Poulsen, 2009; Insel et al., 2010; Feng et al., 2013; Lee and Fung, 2008; Risi et al., 2013; Botysun and Ehlers, 2021). Furthermore, studies of surface uplift of major orogens, such as the Andes, Himalayas, and Tibet, have demonstrated the impacts of topographic evolution on atmospheric circulation patterns and the spatial distribution of $\delta^{18}$O in precipitation ($\delta^{18}$O$_p$) in a more complicated way than how single-site stable isotope paleoaltimetry studies commonly assume (e.g., Takahashi and Battisti, 2007; Yao et al., 2013; Mulch, 2016). This highlights the need for a better quantitative understanding of how topography and regional climate influence the variations of the isotopic composition of ancient waters (Ehlers and Poulsen, 2009; Botsyun et al., 2020; Insel et al., 2012). This study comprises a series of climate model experiments that address this need for the European Alps.

The European Alps have been extensively studied, but only a few studies have addressed the reconstruction of its surface uplift histories with stable isotope paleoaltimetry (e.g., Sharp et al., 2005; Campani et al., 2012; Krsnik et al., 2021). Recent studies



have suggested that the Alps experienced diachronous surface uplift in response to the post-collisional slab break-off and continuing rollback of the lithosphere and its associated mantle delamination (Schlunegger and Kissling, 2018; Handy et al., 2015). Stable isotope-based reconstructions of past surface topography can help constrain such subsurface processes, given that surface elevation is primarily an expression of mantle and lithospheric dynamics. However, whether such geodynamic

processes would yield (spatial) differences in $\delta^{18}O_p$ by a magnitude that would be detectable in the geologic record remains a question to answer. Resolving such a question would justify the use of stable isotope paleoaltimetry reconstructions across the Alps to understand its topographic evolution.

In this study, we simulate changes in regional climate and the oxygen isotopic composition of precipitation ($\delta^{18}O_p$) that would

occur in response to diachronous, along-strike surface uplift variations of the Eastern and West-Central Alps. We address the questions of how much (and where) different scenarios of differentiated west-to-east surface uplift would be reflected in $\delta^{18}O$ of meteoric water. In answering these questions, we test two hypotheses. We hypothesize that different topographic configurations for the Eastern and West-Central Alps result in regional climate and $\delta^{18}O_p$ patterns that are significantly different from (1) those of today and (2) those produced by scenarios of bulk surface uplift of the entire Alps. We test these hypotheses

through a series of sensitivity experiments with two free parameters including variations in the elevation of the West-Central Alps and the elevation of the Eastern Alps. The experiments are conducted with the isotopic-tracking atmospheric GCM ECHAM5-wiso and provide quantitative estimates of the expected $\delta^{18}O_p$ signal that would eventually be recorded in geological archives used in stable isotope paleoaltimetry. The study, therefore, also represents an important step toward answering the question of whether the eastward propagation of surface uplift (or different East-West topographic configurations) would be

detectable in paleo-$\delta^{18}O_p$ records.

## 2 Background

### 2.1 Geodynamics of the European Alps

The European Alps are a mid-latitude orogenic belt extending over a longitudinal area (~ 1000 km²) subdivided into the Western, Central, and Eastern Alps (Schmid et al., 2004). The onset of its topographic development is attributed to the

continent-continent collision of the European and Adriatic plates in the late Eocene. This was followed by protracted convergence and subduction of oceanic lithosphere (Frisch, 1979; McCann, 2008; Schmid et al., 1996; Stampfli et al., 1998). Major rock exhumation started ~35 Ma or earlier, predominantly in response to crustal thickening and associated erosion (Kuhlemann et al., 2002; Schmid et al., 2004; Schmid et al., 1996) and drainage reorganization (Lu et al., 2018). Recent modeling studies have suggested additional geodynamic processes that may have influenced the surface uplift history (Kissling

and Schlunegger, 2018; Schlunegger and Kissling, 2015). These include slab break-off (~ 30 Ma) and slab rollback of the subducting lithosphere, as well as lithospheric mantle removal that may have contributed to west-to-east variations in surface uplift as having been proposed for the Central Alps (Davies and von Blanckenburg, 1995; Schlunegger and Castelltort, 2016;



Ustaszewski et al., 2008). Subsequently, slab break-off (~ 20 Ma) is suggested to have occurred under the Eastern Alps (Handy et al., 2015). Based on the previous studies, current tectonic and geodynamic reconstructions suggest that the Alps did not rise

monotonically but through diachronous surface uplift of different sections of the mountain range. This study considers this scenario and evaluates expected changes in $\delta^{18}O_p$ values that would be preserved in geologic archives within the region.

## 2.2 Paleoaltimetry estimates of the European Alps

Few studies have attempted to reconstruct the surface elevation history of the Alps, and some of these reconstructions contradict geodynamic models. Based on pollen data, Fauquette et al. (2015) estimated that the southwestern Alps reached

their maximum mean elevation of more than ~ 1900 m at the early stage after the collision in the early Oligocene (ca. 30 Ma). Geodynamic modeling and geomorphic analysis suggest that the Eastern Alps initiated their topographic development only in the middle Miocene (~15 - 10 Ma), with a major phase at ~ 5 Ma (Bartosch et al., 2017; Hergarten et al., 2010). However, the estimate of the topographic development of the Eastern Alps is deemed unlikely since significant continental lithosphere subduction under the European plate would have resulted in 200 and 100 km post crustal shortening (and presumably some

thickening) after 40 and 20 Ma, respectively (Lippitsch et al., 2003; Rosenberg et al., 2015). Reconstructions for the Central Alps underline these uncertainties with plausible paleoelevation estimates ranging from 2000 to 6000 m (e.g., Campani et al., 2012; Kocsis et al., 2007; Sharp et al., 2005). Krsnik et al. (2021) recently confirmed the higher end of these previous estimates for the Central Alps, with surface elevations of > 4000 m being attained not later than the middle Miocene. Collectively, these reconstructions suggest that the Western and Central Alps were already at high elevations in the middle Miocene, and surface

uplift must have occurred in the Oligocene to Miocene. In contrast, no long-term quantitative past surface elevation estimates are available for the Eastern Alps. However, if the isotopic signal created by west-to-east surface uplift propagation is preserved and detectable in geological materials such as, e.g., pedogenic carbonates or hydrous shear zone silicates, stable isotope paleoaltimetry may be used to address this research gap.

## 2.3 Climate of the Alps

The interannual and seasonal variability of regional climate in Europe is predominantly controlled by large-scale circulation patterns (Bartolini et al., 2009; Hurrell, 1995). The topography of the Alps greatly impacts mesoscale temperature, precipitation, moisture transport, wind, and other atmospheric elements (Schmidli et al., 2002). The Alps act as an orographic barrier, which affects convective and orographic precipitation formation and its associated spatial effects like leeward rain shadows (Bartolini et al., 2009; Beniston, 2005). Today, the Alps experience maximum precipitation rates in summer due to

(1) the shifting of pressure fronts to the south and (2) high convective heat transport from oceanic sources and continental evapotranspiration (Schmidli et al., 2002). Most atmospheric moisture received across the Alps, especially over the northern flanks, is transported via the westerlies from the North Atlantic. Regional precipitation histories can thus be explained primarily by variations in atmospheric circulation patterns over the North Atlantic and western Europe (Baldini et al., 2008; Comas-Bru et al., 2016; Langebroek et al., 2011; Rozanski et al., 1982). Therefore, $\delta^{18}O$ values in ancient meteoric waters can only be



quantitatively evaluated with knowledge about the dominant large-scale atmospheric flows and the locations of associated
      pressure systems (i.e., the quasi-stable high and low sea level pressure systems), which can shift over time (e.g., Deininger et
      al., 2016).

      Recent advances in climate modeling allow the use of high-resolution isotope tracking Atmospheric General Circulation
      Models (AGCMs) to investigate the impacts of topography and regional climate change on $\delta^{18}O_p$ values (e.g., Botsyun et al.,
2020; Botsyun and Ehlers, 2021; Botsyun et al., 2019; Li et al., 2016; Mutz et al., 2016; Sturm et al., 2010). AGCMs are
      developed based on the recent understanding of atmospheric physics. They can simulate climates that are in dynamic
      equilibrium with prescribed orbital, environmental, and topographic boundary conditions. While GCMs have some
      deficiencies in predicting precipitation in mountain regions due to model-specific parametrization (e.g., cloud microphysics
      and the hydrostatic approximation), they have been shown to adequately reproduce important patterns of climate and
precipitation $\delta^{18}O$ over orogens, including the Alps (e.g., Botsyun et al., 2020; Werner et al., 2011). Previous studies have used
      GCMs to perform topographic sensitivity experiments to help improve paleoaltimetry estimates (e.g., Botsyun et al., 2019;
      Shen and Poulsen, 2019; Ehlers and Poulsen, 2009; Poulsen et al., 2010; Feng et al., 2013; Huyghe et al., 2018; Insel et al.,
      2012). More recently, Botsyun et al. (2020) performed GCM experiments designed to estimate the $\delta^{18}O_p$ response to bulk
      surface elevation changes of the whole Alps. This study builds on their finding by considering diachronous surface uplift
(stepwise surface uplift from west to east) as different topographic scenarios.

## 3 Data and methods

### 3.1 General Circulation Model (ECHAM5-wiso)

The hypotheses tested in this study are addressed with a series of experiments conducted with the isotope-tracking climate
model ECHAM5-wiso. ECHAM5 is the fifth version of a well-established atmospheric GCM that is developed and maintained
by the Max Planck Institute for Meteorology (MPIM) and based on the spectral weather forecast model of the European Centre
      of Medium-Range Weather Forecast (ECMWF) (Roeckner et al., 2003). ECHAM5 has been expanded to include an isotope
      tracking module that simulates the isotopic composition of water at every step of the simulated hydrological cycle in the model
      (Werner et al., 2011). The water isotopologues (i.e., $H_2^{16}O$, $H_2^{18}O$, and HDO) are treated as independent tracers (Hoffmann et
      al., 1998; Werner et al., 2011) that undergo kinetic and equilibrium fractionation during phase transitions in the atmosphere
(e.g., snow, vapor, clouds). The semi-Lagrangian advection scheme is used for the transport of the passive tracers through all
      the water components (Lin and Rood, 1996). The resulting isotope-tracking GCM (ECHAM5-wiso) has been demonstrated to
      reproduce global and regional scale isotopic observations well, including for present-day $\delta^{18}O_p$ values across Europe (Botsyun
      et al., 2020; Werner et al., 2011; Langebroek et al., 2011). Compared to its predecessor, the version applied in this study has
      an improved representation of the land surface and considers orographic drag forces. However, the model does not track
isotopic fractionations from surface waters (Hagemann et al., 2006). The model simulates clouds using the stratiform cloud
      scheme that consists of the prognostic equations of all the water phase dynamics, bulk cloud microphysics by Lorenz and





Lohmann (2004), and statistical cloud cover parametrization by Tompkins (2002). Comprehensive details about the model physics and parameterization are described in Werner et al. (2011) and Roeckner et al. (2003).

**3.2 Topography experiments**

We investigate the effects of specific topographic configurations on $\delta^{18}O_p$ and regional climate by performing sensitivity experiments with two free parameters, (1) the elevation of the West-Central Alps (43 - 48 °N, 5 – 10 °E) and (2) the elevation of the Eastern Alps (45 – 48 °N, 10 – 17 °E). For brevity, a two-part notation is used for individual topographic configurations: The first part denotes the elevation of the Western-Central Alps and assumes the form *Wx*, where *x* expresses the elevation as a fraction of its present-day value. The second part analogously expresses the elevation of the Eastern Alps in the form of *Ex*.

The topographic configuration W2E0, for example, therefore consists of the Western-Central and Eastern Alps set to 200% and 0% of their modern elevation, respectively. The reader is advised that configurations with 0% topography use 250 m as a mean minimum topography to avoid unrealistic artifacts in the simulations, such as extreme wind speeds due to a completely flat low-elevation surface.

We elaborate on two topographic scenarios, each consisting of several topographic configurations (see Table 1 for a complete overview).

Scenario 1: The first scenario considers the diachronous west-to-east surface uplift hypothesized from tectonic reconstructions of the Alps (e.g., Bartosch et al., 2017; Fauquette et al., 2015; Handy et al., 2015). Consequently, the West-Central and Eastern Alps are varied separately. First, the elevation of the Western-Central Alps was kept at its present-day value (W1), and the

elevation of the Eastern Alps was incrementally increased from 0% to 200% of its present-day value. Following this, the West-Central Alps were set to 200% (W2) of their present-day elevation, and the Eastern Alps were raised incrementally again. W2 was chosen to represent a plausible middle Miocene altitude of more than 4000 m, which is close to 200% of the modern mean elevation of the Central Alps (Krsnik et al., 2021). However, the resolution of the model underrepresents the magnitude of the orographic mean elevation due to dampening of maximum peak elevations across the Alps in the interpolation process.

Scenario 2: For the second scenario, the topography of the entire Alps, including the West-Central and Eastern Alps, was increased to 200% (W2E2) and reduced to 0% (W0E0) of its present-day height. These topographic configurations allow for comparing the climatic responses to diachronous surface uplift (scenario 1) and bulk surface uplift. Since the W1E1 configuration simply represents modern topography, it is used as the control experiment and given the special designation CTL.






| Model set-up | Experiment name | Topography configuration | Boundary Conditions |
|---|---|---|---|
| **Present-day (1979-2000)** | PD | 100% of the present-day elevation | Present-day (e.g. $CO_2$ = 348 ppm) |
| **Control** | CTL | 100% of the present-day elevation | Pre-industrial (e.g. $CO_2$ = 280 ppm) |
| **Scenario 2 (bulk topographic change)** | W2E2 | 200 % of the present-day elevation | same as CTL |
| | W0E0 | The Alps reduced to 250 m | same as CTL |
| **Scenario 1 (W1)** | W1E0 | 100 % of present-day Western-Central Alps elevation and reduction of Eastern Alps to 250 m | same as CTL |
| | W1E1.5 | 100 % of present-day Western-Central Alps elevation and 150 % of present-day Eastern Alps elevation | same as CTL |
| | W1E2 | 100 % of present-day Western-Central Alps elevation and 200% % of present-day Eastern Alps elevation | same as CTL |
| **Scenario 1 (W2)** | W2E0 | 200 % of present-day Western-Central Alps elevation and reduction of present-day Eastern Alps elevation to 250 m | same as CTL |
| | W2W1 | 200 % of present-day Western-Central Alps elevation and 100 % of present-day Eastern Alps elevation | same as CTL |

**Table 1: Summary of ECHAM5-wiso sensitivity experiments and their topographic configurations and boundary conditions.**

### 3.3 Model setup and boundary conditions

All experiments were performed for 18 model years on a high spatial resolution grid to represent the Alps' topography adequately. Specifically, the T159 spectral resolution (which corresponds to ~0.75° or ~80 km in latitude and longitude) and

31 vertical pressure levels (up to 10 hPa) were used for the simulations. The output frequency was set to 6 model hours to allow the performance of a trajectory analysis (section 3.6). We only consider the last 15 years of the model output and remove the first three years of the simulation to account for the spin-up period, i.e., the time needed for the simulated climate to reach a dynamic equilibrium. Since this study aims to quantify and isolate the effects of different topographic configurations (section 3.2) on regional climate and $\delta^{18}O_p$, the values for all other boundary conditions are kept constant at pre-industrial (PI) levels.

These include orbital configurations, greenhouse gas concentrations, sea surface variables, and insolation. For model validation purposes, we additionally conduct a present-day (PD) simulation of 43 model years and analyze the last 30 years (1979-2000).





We use the annual mean variations of the sea surface temperature (SSTs) and sea ice concentrations (SICs) from the Atmospheric Model Intercomparison Project (AMIP) as boundary conditions. The simulated PD climate and isotopic patterns are compared with observed values across Europe. The reader is referred to Mutz et al. (2016) and (2018) for more details about the PD and PI boundary conditions used in this study.

### 3.4 Model-Data comparison

Modern station-based $\delta^{18}O_p$ data from the Global Network of Isotopes in Precipitation (GNIP) in Europe (accessible at https://www.iaea.org) is used as our first validation dataset (Edwards et al., 2002). The precipitation weighted $\delta^{18}O_p$ values from GNIP stations are compared to the PD simulation to assess the performance of ECHAM5-wiso. Specifically, we compute and compare long-term annual means of precipitation weighted $\delta^{18}O_p$ for the period covered by the GNIP station measurements across the European continent. The ERA5 climate reanalysis, produced and managed by the ECMWF, is our second validation dataset. The reanalysis is a state-of-the-art, globally gridded dataset produced from both physical models and observations (e.g., ocean buoys, aircraft, and other platforms) that is dynamically interpolated using the four-dimensional variational (4D-Var) data assimilation scheme (Hersbach et al., 2020). Compared to its predecessor ERA-Interim (Dee et al., 2011), the dataset has improved in both temporal (hourly throughout) and spatial resolution (31 km or TL639) and expanded its data coverage from 1950 onwards (Bell et al., 2021). We use this dataset to construct Northern Hemisphere teleconnection patterns and compare them to their equivalents constructed from topographic scenarios simulations.

### 3.5 Postprocessing and analysis of simulation

Long-term seasonal and annual arithmetic means were calculated from the 6h model output. The deviations of these means from the CTL mean are calculated by subtracting the CTL mean from the topography scenarios. The resulting anomalies are referred to as "ID - CTL mean differences" hereafter, where ID is an experiment ID such as, e.g., W2E1 (see section 3.2 and Table 1 for an overview). Two-tailed student t-tests with a defined confidence interval threshold of 95% were applied to assess the statistical significance of these differences. In this study, we mainly discuss summer (JJA) estimates since numerous studies use pedogenic carbonates, which are preferentially formed during soil drying when evaporation exceeds precipitation (e.g., Gallagher et al., 2019; Breecker et al., 2009; Zamanian et al., 2016) as a proxy for stable isotope paleoaltimetry. However, since there are uncertainties about the extent of the seasonal bias in pedogenic carbonate formation (e.g., Kelson et al., 2020), the annual means are also provided. The elevation-$\delta^{18}O_p$ relationships, further referred to as the isotopic lapse rates (ILRs), were estimated for different geographic areas around the Alps (Fig. 1 A) by performing Ordinary Least Squares (OLS) linear regressions on the grid point values within each region. We use the notation -1‰/km (instead of 1‰/km) to report a decrease of 1‰ for an elevation increase of 1km.

The prominent Northern Hemisphere teleconnection patterns (i.e., North Atlantic Oscillation (NAO), East Atlantic (EA), Scandinavian (SCAN), and East Atlantic/Western Russia (EA/WR) patterns) were extracted from the model output to investigate the influence of surface uplift on synoptic-scale atmospheric variability, which in turn affects atmospheric moisture





transport and $\delta^{18}O_p$ values. These were captured by conducting a Principal Component Analysis (PCA) or Empirical
Orthogonal Function (EOF) analysis (e.g., von Storch and Zwiers, 2001; Hannachi et al., 2007) on the summer (JJA) mean sea
level pressure (slp) fields in the North Atlantic-European domain (20 - 80 °N, 80 °W - 40 °E). Before the EOF analysis, a
spatial weighting on the latitude of the pressure anomalies was applied to equalize the atmospheric field geographically, as
North et al. (1982) recommended. The patterns extracted from present-day ERA5 mean sea level pressure data were used as a
reference to help group the modes of variability from the topography experiments.

For further analysis of the effects of changing topography on regional climate across the orogen, we extracted variations in
vertical wind velocity (omega), cloud cover, and relative humidity along-strike (west-to-east) of the Alps. Since the position
of the Alps is approximately parallel to the present-day prevailing wind direction, along-strike variations in these climate
elements can provide insight into the potential evolution of air parcels originating from the west. This analysis is complemented
by trajectory analyses (see section 3.6).

**3.6 Trajectory analysis**

Kinetic back trajectory analyses were performed to investigate the impacts of the topography scenarios on moisture source and
transport across the Alps. Specifically, the Lagrangian analysis tool "LAGRANTO" (Sprenger and Wernli, 2015) was used
with the three-dimensional wind fields (i.e., zonal (u), meridional (v), vertical (omega) wind velocities) of the 6h model outputs
for the trajectory analysis. The tool uses a robust numerical scheme with efficient spatial interpolation (bilinear and linear
interpolation for the horizontal and vertical directions, respectively). The trajectories were back-tracked for 5 days from a
receptor point at the 850 hPa vertical level defined at four different locations (i.e., Graz (47.06 °N, 15.44 °E), Munich (48.14
°N, 11.53 °E), Bologna (44.49 °N, 11.38°E) and Lyon (45.81 °N, 4.82 °E)).

**4 Results**

This section summarizes the simulated changes in regional climate and $\delta^{18}O_p$ in response to the different topographic
configurations across the Alps (section 3.2, Fig. S1). More specifically, the presentation of results focuses on $\delta^{18}O_p$ values,
isotopic lapse rate, precipitation amount, near-surface temperature, moisture transport, and atmospheric circulation patterns.
The section comprises present-day model validation with observed $\delta^{18}O_p$ (section 4.1), a summary of the CTL experiment
(section 4.2), and the changes mentioned above relative to the CTL simulation (section 4.3 - 4.10). Unless stated otherwise,
the results are presented for the summer (JJA) season. The annual scale changes are included in the supplementary material to
this manuscript.

**4.1 Present-day (PD) simulation and model validation**

The simulated annual mean $\delta^{18}O_p$ values decrease from the North Atlantic Ocean towards eastern Europe and over the Alps by
10 to 12 ‰ and deviate from the observed GNIP data only slightly by 1-2 ‰ (Fig. 1 B). The annual means of near-surface





temperature shows 0 - 4 °C across the Alps and precipitation of ∼ 150 - 200 mm/month, which is higher than its adjacent low-
elevation regions. Low-level winds originating from the North Atlantic travel toward Europe and show slight deflections across
the Alps (Fig. 1 D). Overall, comparing the annual long-term means of the model outputs to observed GNIP stations $\delta^{18}O_p$
values and observed PD precipitation and temperature patterns indicates that the model reasonably represents $\delta^{18}O_p$ values and
the regional climate across Europe. The topography used as an input parameter for the model moderately represents the
topography of the Alps with a dampened elevation of the highest peaks (Fig. 1 A).




**Figure 1: Present-day (PD) simulation topography (with geographical windows used for isotopic lapse rate calculations) (A),
simulated (shading on map), and observed (circles) annual means of δ¹⁸Oₚ values. Colored circles represent observed δ¹⁸Oₚ values
obtained from GNIP network stations (B), simulated near-surface temperature (C), and precipitation amount (colored shading) with
near-surface wind patterns (arrows: length of arrows indicates wind speed, m/s) (D).**





### 4.2 Control simulation (CTL) $\delta^{18}O_p$ values, near-surface temperature, and precipitation

The simulation with pre-industrial (PI) boundary conditions and PD topography did not show a significant difference in predicted $\delta^{18}O_p$ values compared to the PD simulation. Overall, the CTL experiment shows decreasing patterns of summer $\delta^{18}O_p$ values towards north-eastern Europe and predicts $^{18}O$-depleted $\delta^{18}O_p$ values in the range of -10 to -12 ‰ across the Alps

(Fig. 2 A). However, the $\delta^{18}O_p$ values slightly increase towards the east of the Alps (~ 20 °E), yielding values from -10 to -6 ‰. Near-surface temperatures are estimated to be more than 10 °C across Europe, with minimum values across the Alps (2 - 6 °C) and a cooling gradient towards the north (Fig. 3 A). Moreover, the predicted patterns show maximum near-surface temperatures across the Mediterranean region. The simulated precipitation patterns decrease towards the south, with the Mediterranean region having the driest climate (Fig. 4 A). However, the topography of the Alps coincides with higher

precipitation (an average of summer months of ~ 125 mm/month) compared to estimates across Central Europe. On the western flanks of the Alps, low-level westerly winds are deflected in a north-south bifurcation pattern (Fig. 4 A).

### 4.3 Changes in $\delta^{18}O_p$ values for the different topographic scenarios

The simulated regional patterns of $\delta^{18}O_p$ values show significant changes in response to the different topography scenarios, especially in regions of modified topography. Overall, the $\delta^{18}O_p$ values decrease with increasing elevation. This decrease is

accentuated when the topography of the West-Central Alps is higher (i.e., for the W2 configurations of scenario 1). More specifically, the W2E1 experiment predicts a significant decrease in $\delta^{18}O_p$ values across the Alps in the range of 2 to 8 ‰ with minimum $\delta^{18}O_p$ values in the West-Central Alps (Fig. 2 B). The W2E0 experiment predicts a more localized significant decrease of 2 to 6 ‰ in the West-Central Alps (Fig. 2 D). In contrast, the W1E2 configuration results in a decrease of 2 to 6 ‰ over the extended east flank of the Alps (Fig. 2 E), and the W1E0 simulation does not predict any statistically significant

changes in the orogen (Fig. 2 C). The W2E2 configuration (of scenario 2) results in a decrease of $\delta^{18}O_p$ values in the range of 2 to 6 ‰ across the Alps and its adjacent low-elevation regions (Fig. 2 F). However, the W2E2 experiment also predicts a substantial increase in $\delta^{18}O_p$ values across northwest Europe (i.e., over Ireland and the UK). All experiments predict changes in $\delta^{18}O_p$ values of 1 to 2 ‰ across Europe independent of direct topographic changes. However, these independent changes are only statistically significant for the W2 experiments. On the annual scale, the predictions show similar patterns, but changes

are more localized and reduced in magnitude by ~ 2 ‰ (Fig. S5).





**Figure 2: Seasonal (JJA) long-term mean of δ¹⁸Oₚ values of the CTL experiment (A) and seasonal (JJA) mean difference of δ¹⁸Oₚ values for the different topography scenarios (i.e., W2E1-CTL (B), W1E0-CTL (C), W2E0-CTL (D), W1E2-CTL (E) and W2E2-CTL (F)). Red color ranges represent heavy isotope depletion, and blue color ranges represent an enrichment in heavy isotopes relative to the CTL experiment. Regions that experience changes that are statistically significant, as indicated by student t-test analysis with a 95% confidence level, are marked with black slash stippling.**

## 4.4 Changes in near-surface temperature for the topographic scenarios

The topographic scenarios predict significant localized cooling (warming) where the topography is raised (lowered). The W2E1 experiment predicts a significant decrease of 5 to 12 °C in the West-Central Alps (Fig. 3 B). The W2E0 simulation predicts similar changes in the West-Central Alps, but with a corresponding increase of ~ 5 °C in the Eastern Alps in response to the reduced elevation (Fig. 3 D). On the other hand, the W1E2 simulation predicts a significant decrease of near-surface temperature by 2 to 7 °C in the Eastern Alps and shows a slight increase of ~ 1 °C over southeast Europe (Fig. 3 E). The W1E0





experiment also estimates an increase of 2 to 5 °C in the Eastern Alps (Fig. 3 C). In total, the topographic configurations with
a steeper gradient across the Alps (e.g., W2E0 and W1E0) result in a decrease in near-surface temperatures of ~ 1 °C in parts
of northeast Europe. The W2E2 configuration (of scenario 2) results in a significant temperature decrease of 5 to 10 °C across
the Alps from west to east and shows larger affected low-elevation areas around the Alps (Fig. 3 F). Specifically, the changes
show a decreasing pattern towards the eastern flank of the Alps. The simulated patterns on the annual scale are very similar to
the summer estimates (Fig. S6).

**Figure 3: Seasonal (JJA) long-term mean of the near-surface temperature of the CTL experiment (A) and seasonal (JJA) mean
difference of near-surface temperature for the different topography scenarios (i.e., W2E1-CTL (B), W1E0-CTL (C), W2E0-CTL
(D), W1E2-CTL (E) and W2E2-CTL (F)). Red color ranges represent warmer, and blue color ranges represent colder temperatures
than in the CTL experiment. Regions that experience changes that are statistically significant, as indicated by student t-test analysis
with a 95% confidence level, are marked with black slash stippling.**



**4.5 Changes in precipitation for the topographic scenarios**

For all scenarios, an increase in elevation results in an increase in precipitation across the orogen. The W2 experiments of
scenario 1 predict more important changes in the orogen and other parts of Europe than the W1 experiments. The W2E1
experiment predicts a significant increase in precipitation of up to 125 mm/month across the Alps and a decrease of 25
mm/month over eastern and central Europe around the Alps (Fig. 4 B). Moreover, the W2E0 experiment predicts an increase
of up to 100 mm/month in the West-Central Alps and shows a more widespread decrease in the surrounding regions (Fig. 4

D). In contrast, the W1E2 experiment estimates an increase of < 80 mm/month across the Alps from the west, which peaks in
the Eastern Alps (Fig. 4 E), and the W1E0 experiment predicts a decrease of ~ 25 mm/month in the Eastern Alps (Fig. 4 C).
The W2E2 configuration of scenario 2 results in a significant increase of up to 125 mm/month across the Alps from west to
east and a decrease toward northern and eastern Europe (Fig. 4 F). However, only precipitation increases in northwest Russia
and northern Europe for experiments W2E1 and W2E0, respectively, are notable and statistically significant changes far from

the orogen. Over annual timescales, changes are more localized and restricted to regions with modified topography (Fig. S7).



**Figure 4: Seasonal (JJA) long-term mean of precipitation amount(shading) with near-surface wind patterns (arrows) of the CTL experiment (A) and mean difference of precipitation amount for the different topography scenarios (i.e., W2E1-CTL (B), W1E0-CTL (C), W2E0-CTL (D), W1E2-CTL (E) and W2E2-CTL (F)). Green color ranges represent wetter, and brown color ranges represent drier conditions than in the CTL experiment. Regions that experience changes that are statistically significant, as indicated by student t-test analysis with a 95% confidence level, are marked with black slash stippling.**

## 4.6 Spatial profiles of $\delta^{18}O_p$ values across the Alps

Spatial mean oxygen isotopic profiles across the Alps in the longitudinal (46 – 47 °N) and latitudinal (11 – 15 °E) directions reveal varied responses to the different topographic scenarios. Overall, the isotopic profiles show a decrease in $\delta^{18}O_p$ values across the Alps from west to east and from south to north. The difference in $\delta^{18}O_p$ values along the profiles is estimated to be less than -2 ‰ in low-elevation regions adjacent to the mountains and up to -8 ‰ in modified topography regions (Fig. 5). The





W1E2, W1E1.5, and W1E0 configurations of scenario 1 result in locally low $\delta^{18}O_p$ values down to -10 ‰, -8 ‰ and -5 ‰ in

the Eastern Alps, respectively (Fig. 5 A). However, the W1 experiments predict no significant changes in the West-Central Alps and the north-south direction. In contrast, the W2E1 and W2E0 experiments of scenario 1 predict a decrease down to -14 ‰ in the West-Central Alps. The isotopic values gradually increase up to -8 ‰ and -6 ‰ in the Eastern Alps for the W2E1 and W2E0, respectively.

The comparison of the isotopic profiles for scenarios 1 and 2 (i.e., between the diachronous and bulk surface uplift experiments)

reveals a significant difference along the strike of the Alps. The differences in response to the topographic forcing of both scenarios are more visible across the Eastern Alps. They are greater by -0.5 to -2 ‰ for scenario 2 (Fig. S8 A). However, the scenario 2 experiments show less negative $\delta^{18}O_p$ values across the West-Central Alps (Fig. S8 B).




**Figure 5: Regional seasonal (JJA) means of spatial δ¹⁸Oₚ values across the Alps in longitudinal (averaged between 46°N and 47°N) (A, C) and latitudinal (averaged between 11°E and 15°E) (B, D) direction in response to different topographic scenarios (CTL (black), W1E0 (blue), W1E2 (red), W1E1.5 (green), W2E1 (gold) and W2E2 (purple). Grey shadings represent topography profiles of W1 (A, B) and W2 (C, D) scenario 1 experiment.**

## 4.7 Changes in isotopic lapse rate in response to different scenarios

The isotopic lapse rates (ILRs) estimated for the different geographical windows around the Alps (Fig. 1 A) show varied responses to the different topographic configurations. In relation to CTL experiment ILR estimates (i.e., -2.31, -2.18, and -3.11 ‰/km for the west, north, and south transects, respectively), the W1 experiments predict a decrease in ILRs for the western and southern transects (Fig. 6), and the W2 experiments predict an increase in the west and north transects (Fig. 7). Note that the W1E0 experiment estimates a dampened ILR due to the simultaneous increase in δ¹⁸Oₚ values of the low elevation areas (i.e., -2.08 ‰/km for the west, -2.01 ‰/km for the north, and -2.17 ‰/km for the south transect) (Fig. 6). The W1E2 simulation predicts steeper ILR for the north (-2.61 ‰/km) and a shallower ILR for the west (-1.83 ‰/km) and south (-2.96 ‰/km) transects (Fig. 6). The W2E1 and W2E0 experiments predict steeper ILRs for both the western (-2.78 and -2.68 ‰/km) and northern (-3.37 and -3.22 ‰/km) flanks and a dampened ILR for the south (-2.91 and -2.88 ‰/km) transect (Fig. 7). On the other hand, the W2E2 experiment estimated a shallower ILR of -1.49 ‰/km for the west and -2.39 ‰/km for the south transect, but a steeper ILR of -2.59 ‰/km for the north transect (Fig. 7). The r² values associated with the ILRs exceeded 0.85. The estimated ILR changes using annual means are comparable to the patterns of summer means but with generally steeper gradients (cf. Fig. S9 and S10 in the supplemental material).



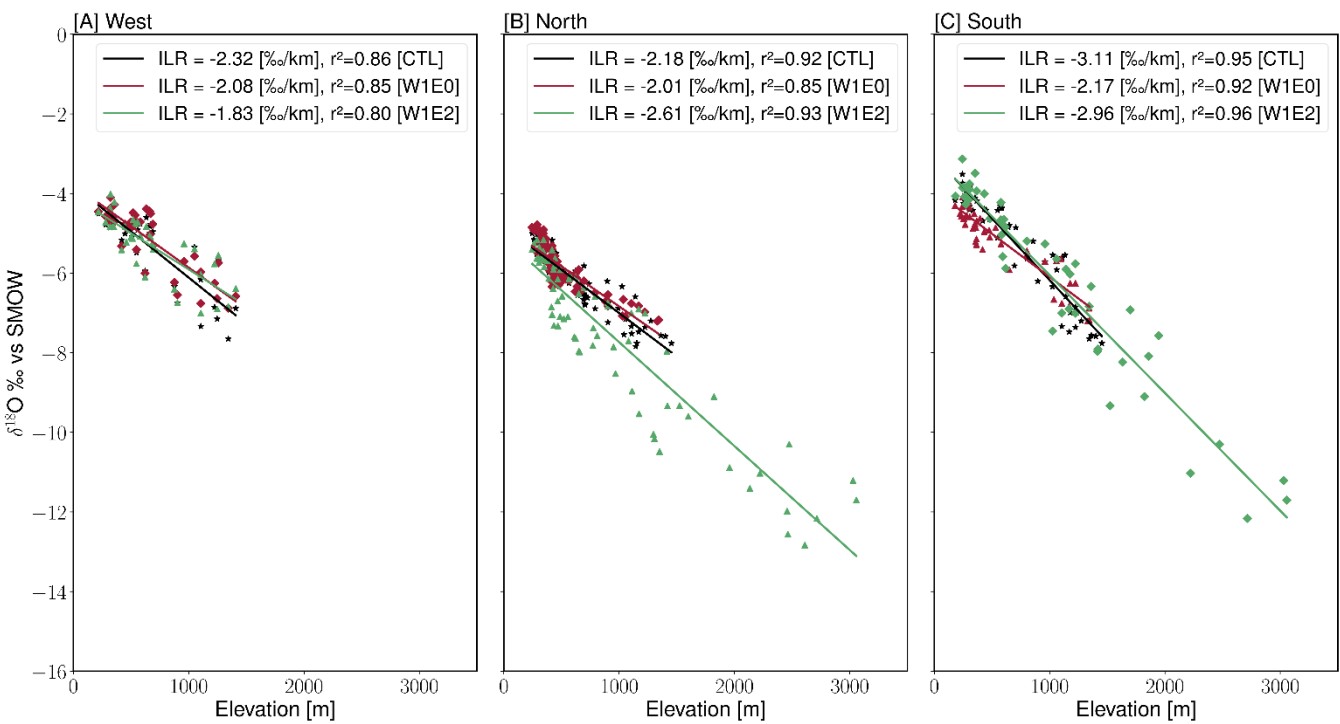

**Figure 6: Summer isotopic lapse rates (ILRs) estimates for the W1 topography scenarios (i.e., W1E0 (red), W1E2 (green)), and CTL (black) experiments for the different transects around the Alps as shown in Fig. 1 A (West: 44 - 47 °N, 1- 8 °E, south: 43 - 47 °N, 8 - 15 °E, and north: 47 - 50 °N, 5-16 °E). The ILRs are estimated as the δ¹⁸Oₚ-elevation gradients using linear regression. r² values represent the associated correlation coefficients.**





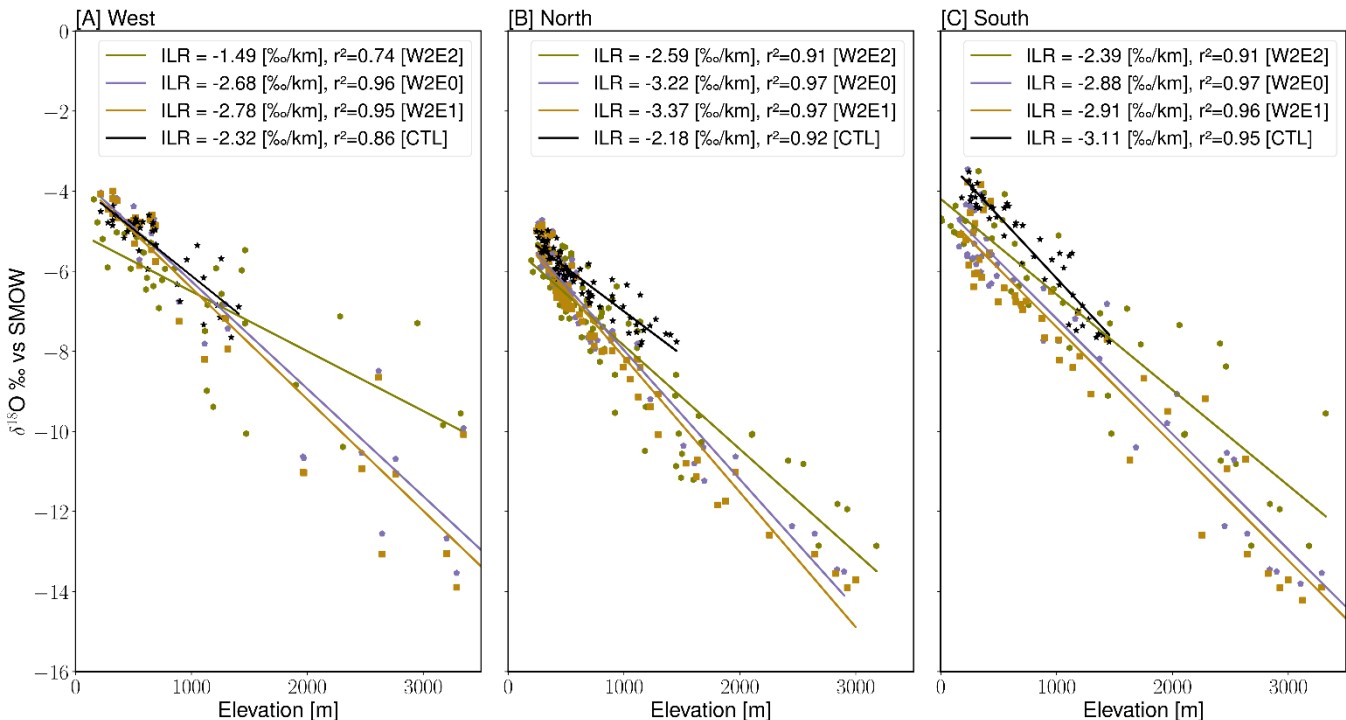

**Figure 7: Summer isotopic lapse rates (ILR) estimates for the W2 topography scenarios (i.e., W2E0 (purple), W2E1 (gold)), CTL (black), and W2E2 (olive) experiments for the different transects around the Alps as shown in Fig. 1 A ((West: 44 - 47 °N, 1- 8 °E, south: 43 - 47 °N, 8 -15 °E, and north: 47 - 50 °N, 5-16 °E). The ILRs are estimated as the $\delta^{18}O_p$-elevation gradients using linear regression. $r^2$ values represent the associated correlation coefficients.**

### 4.8 Changes in moisture source and transport

The back-trajectory analyses demonstrate that the topographic scenarios have a significant influence on air (and therefore moisture) transport towards target regions on the eastern and southern flanks of the Alps. Therefore, the presentation of results (and the associated discussion in section 5) focuses on regions around the cities of Graz (Fig. 8) and Bologna (Fig. 9), which represent locations in the east and south, respectively. The reader is referred to the supplemental material for the back-trajectories for the regions around Lyon (in the west) and Munich (to the north) (Fig. S11 and S12).

Overall, the model tracks most summer air masses back to the North Atlantic and some to a continental moisture source in Western Europe, depending on the topography scenario and target location. The W2 experiments of scenario 1 (i.e., W2E1 and W2E0) show moisture ascending over the West-Central Alps on a higher vertical level before finally descending to the target location in Graz (Fig. 8 B and D). Moreover, the W2 experiments also show slight moisture sources from the southern flank of the Alps and predict a shorter moisture transport distance from the North Atlantic relative to the CTL experiment. However, the W1E0 experiment trajectories deviate slightly from the CTL experiment (Fig. 8 C). Overall, the W1E2 and W2E2 trajectories toward Graz (Fig. 8 E and F) show significant deviations from the CTL trajectories (Fig. 8 A). The W1E2 and W2E2 back-track low-level air masses over northwest Europe, showing a gradual ascent over the Alps towards the east at



a shorter distance. However, the W1E2 trajectories show air mass transport on the higher level (~ 700 hPa) directly from the

North Atlantic to the target region on the east flank of the Alps without any orographic barrier deflection (Fig. 8 E).

**Figure 8: 5 days summer back-trajectories with the receptor location set to the 850 hPa level above Graz (47.06 °N, 15.44 °E). The colored lines represent the vertical pressure level of the trajectories. The trajectories were estimated with the 6h wind fields (i.e., u, v, and omega) from the topographic experiments using the LAGRANTO tool.**


For the CTL experiment, the air mass transport and distance to the southern location (i.e., Bologna, Fig. 9 A) are similar to the

results for Graz, but the moisture originates from a higher atmospheric level. The W2 experiments of scenario 1 also predict a

significant influence on air mass trajectories to the southern flanks of the Alps. Specifically, the air masses from experiments

W2E1 and W2E0 originate from the North Atlantic at a higher atmospheric level (~ 750 hPa or less) and then divert towards





the south-east at the western flank of the Alps before being transported to Bologna (Fig. 9 B, and D). Moreover, part of the air mass is transported across the Northern Alps and then diverted downwards through the eastern flank to the receptor location in the south. The W2E0 experiment shows a shorter moisture transport distance than the W2E1 trajectories. For the W1 experiments, the W1E0 shows no significant difference in moisture transport compared to the CTL trajectories (Fig. 9 C). The W1E2 predicts a longer moisture transport distance from the North Atlantic at a higher atmospheric level and an ascent over

the Alps towards the southern flanks (Fig. 9 E). On the other hand, the results from the W2E2 experiment show air masses from Western Europe descending towards the south (Fig. 9 F). Moreover, the W2E2 experiment also shows some trajectories from the eastern side of the Alps for the calculated 5-day back-trajectories. Overall, the target regions at the north and west flanks show fewer significant changes in air mass transport and source in response to the different topographic forcings (Fig. S11 and S12).




**Figure 9: 5 days summer back-trajectories with the receptor location set to the 850 hPa level above Bologna (44.49 °N, 11.38 °E). The colored lines represent the vertical pressure level of the trajectories. The trajectories were estimated with the 6h wind fields (i.e., u, v, and omega) from the topographic experiments using the LAGRANTO tool.**

## 4.9 Vertical structure of vertical wind velocity, cloud cover, and relative humidity across the Alps

The vertical cross-sections of the Alps reveal important changes in the tropospheric climate structure in response to the different topographic configurations. The CTL experiment shows negative omega values, indicating wind directions away from the ground, up to the ~ 600 hPa atmospheric level in the West-Central Alps, and positive omega values towards the Eastern Alps (Fig. 10 A). These regions of updraft (and subsidence) coincide with regions of high (and low) cloud formation (Fig. 10 D). Moreover, the CTL experiment predicts a general decrease in relative humidity from low to high altitude levels

(and from west to east) across the Alps but also predicts more humidity near the tropopause (Fig. 10 G). Overall, the W1E0 experiment simulates a similar atmospheric structure in the West-Central Alps (Fig. 10 B, E, and H). However, the W1E2 atmospheric structure shows alternating moisture ascent and subsidence across the Alps. More specifically, another area of ascent is introduced over the elevated Eastern Alps peak (Fig. 10 C). Cloud formation and relative humidity mimic this pattern, with high cloud cover and high relative humidity coinciding with negative omega (Fig. 10 E and I). The W2 experiments show

a more significant influence on the vertical atmospheric structure than the W1 experiments, especially at the upper-tropospheric levels (Fig. 11). The W2E1 and W2E0 cross-sections show a strong ascent velocity in the West-Central Alps up to the upper troposphere, and low-level subsidence in the Eastern Alps (Fig. 11 A, and B). A vertically extended region of strong cloud formation and high relative humidity spatially coincides with the area of ascent over the West-Central Alps (Fig. 11 D, E, G, and H). The high topography in scenario 2 (W2E2) results in an alternating pattern of positive and negative omega values that

correspond to topographic troughs and peaks, respectively (Fig. 11 C). Ascent (subsidence) spatially coincides with strong (weak) cloud formation and high (low) relative humidity (Fig. 11 F, and I).





**Figure 10:** Seasonal (JJA) means of vertical wind velocity (omega) (A-C), cloud cover (D-F), and relative humidity (G-I) averaged between 45 - 48 °N across the Alps for the W1 (i.e., W1E0, W1E2) and CTL experiments. The black shadings represent the cross-section of topography for each scenario. The omega values represent the speed of air motion in the upward or downward direction. Since vertical pressure decreases with height, negative values indicate upward or ascent velocity and positive values indicate downward or subsidence velocity.

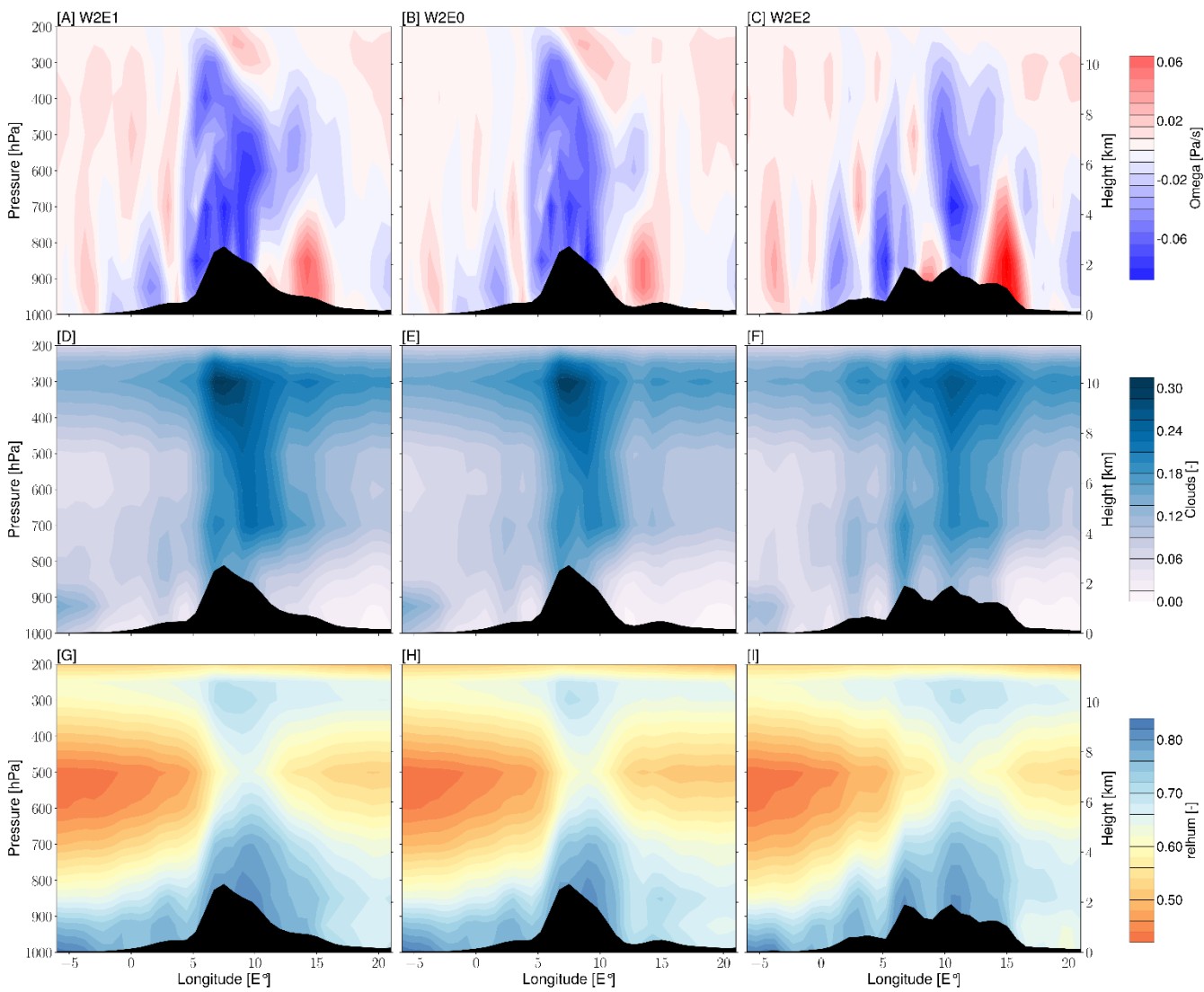

**Figure 11: Seasonal (JJA) means of vertical wind velocity (omega) (A-C), cloud cover (D-F), and relative humidity (G-I) averaged**
**between 45 - 48 °N across the Alps for the W2 of scenario 1 (i.e., W1E0, W1E2), scenario 2 (i.e., W2E2) and CTL experiments. The black shadings represent the cross-section of topography for each scenario. The omega values represent the speed of air motion in the upward or downward direction. Since vertical pressure decreases with height, negative values indicate upward, or ascent velocity and positive values indicate downward or subsidence velocity.**

**4.10 Changes in atmospheric pressure systems in response to different topographic scenarios**

Overall, the different topographic configurations explored in this study impact Northern Hemisphere atmospheric teleconnection patterns. These impacts include geographical shifts and changes in the intensity of quasi-stationary pressure systems (centers of action). The leading mode of atmospheric pressure variability, determined by Empirical Orthogonal Function (EOF) analyses, explains 22 - 35 % of the total pressure variance and is characterized by negative anomalies across Iceland and Greenland, a strong positive anomaly in the mid-latitude North Atlantic Ocean near the Azores, and a weak positive



anomaly over mid-latitude continental Europe. (Fig. 12). These north-south dipole patterns are comparable to the leading mode
        constructed from ERA5 data (cf. Fig. S13 A, supplemental material). The topographic configurations with high elevation
        gradients (i.e., W1E0 and W2E0) result in maximum positive anomalies over continental Europe (Fig. 12 B and D). The W1E2
        experiment shifts the main centers of action of the positive anomaly eastwards by ~20°E and the nodal line of the dipole axis
        (i.e., the line that separates the positive and negative pressure anomalies) northward by ~10°N (Fig. 12 C). The W2E1 and

W2E2 show two well-defined regions of maximum pressure positive anomalies over the North Atlantic and northeast Europe
        (Fig. 12 E and F). In summary, topographic configurations with high West-Central Alps topography (i.e., W1E0, W2E0,
        W2E1) result in an intensification of positive anomalies over continental Europe, while experiments forced with higher Eastern
        Alps topography (i.e., W1E2) shift the maximum positive anomalies region to East Atlantic (Fig. 12 C).

**Figure 12: The spatial patterns and explained variance of the first Empirical Orthogonal Function (EOFs) extracted from the
        topography experiments. These resemble the leading mode of variability extracted from the ERA5 data (cf. Fig. S13 in the**



**supplemental material) and the patterns associated with the North Atlantic Oscillation. The patterns are calculated from summer slp anomalies and represent the covariance matrix of the Principal Component time series and the EOFs.**

The pressure systems with the second mode of variability show a monopole pattern of positive anomalies spread across
northern (> 50 °N) Europe for most of the topographic configurations. This is comparable to the second EOF pattern extracted
from ERA5 data (Fig. S13 B). The CTL experiment shows two well-defined regions of maximum positive anomalies over the
North Atlantic, western Russia, and the Baltic states (Fig. 13 A). The W1E0 simulation predicts a similar spatial pattern as the
CTL experiment but intensifies the eastern anomaly (Fig. 13 B). The W1E2 and W2E0 configurations significantly intensify
the positive anomalies over the East Atlantic (Fig 13 C and D). The W2E1 and W2E2 experiments predict a slight (~ 5°)
northward shift of the band of positive anomalies over the eastern Atlantic Ocean and western Europe. Additionally, the W2E2
experiment also results in a northward shift of positive anomalies in eastern Europe. The spatial patterns of the 3rd and 4th
EOFs are similar to the ERA5 patterns but are not as clearly defined or sensitive to changes in topographic forcing (Fig. S14
and S15).





**Figure 13: The spatial patterns and explained variance of the Empirical Orthogonal Functions (EOFs) that resemble the second mode of variability extracted from the ERA5 data (cf. Fig. S13 in the supplemental material). The patterns are calculated from summer slp anomalies and represent the covariance matrix of the Principal Component time series and the EOFs.**

## 5 Discussion

In the following sections, the impacts of diachronous surface uplift on regional climate (e.g., near-surface temperature, precipitation dynamics, moisture transport, low-level circulation patterns) and their implications on $\delta^{18}O_p$ and isotopic lapse rates (ILRs) are discussed. While the $\delta^{18}O_p$ response is created by the combined effects of all the climate variables previously discussed, we discuss the individual impacts to disentangle the total signal and explain plausible processes for each signal component. Finally, we highlight the study's limitations and implications for stable isotope paleoaltimetry across the Alps (sections 5.6 and 5.7).



**5.1 Impacts of diachronous surface uplift on near-surface temperatures**

The topography sensitivity experiments show significant localized changes in the near-surface temperature. For all topographic configurations, maximum changes were estimated for regions of modified topography, while changes in regions farther from the orogen are less pronounced. In principle, the localized near-surface temperature is a function of the adiabatic temperature lapse rate, which is estimated to vary in the range of 4.1 to 5.9 °C/km across the Alps (Kirchner et al., 2009; Rolland, 2003). Therefore, if the mean of the plausible lapse rates (~ 5 °C/km) is considered, the W2E1 experiment, which represents surface elevation changes of 2 km (Fig. S1) in the West-Central Alps, would predict an adiabatic temperature decrease of ~ 10 °C. The simulated decrease of ~ 12 °C is close to this approximation (Fig. 3 B), suggesting that a large part (ca. more than 80%) of the modeled temperature change can be attributed to the regional temperature lapse rate (i.e., changes in surface elevation), and a relatively small part (~ 2°C) may be due to non-adiabatic processes (i.e., regional climate change). However, given the large range of plausible temperature lapse rate estimates, it is also reasonable to attribute ~80% (in the range of 68 - 98 % from the 4.1 - 5.9 °C/km, respectively) of the temperature changes to simple adiabatic changes. Previous studies on major mountain ranges such as the Andean Plateau also predict additional non-adiabatic temperature changes in response to surface uplift (Ehlers and Poulsen, 2009). The remainder of the signal may be explained by non-adiabatic, climate change-related processes such as changes in tropospheric dynamics, local atmospheric humidity, and atmospheric circulation patterns (sections 5.3 and 5.4). In summary, our results suggest that the simulated temperature differences are produced by adiabatic cooling or heating, with a small contribution from non-adiabatic processes.

**5.2 Impacts of diachronous surface uplift on precipitation**

Topography affects the environment in which precipitation occurs in response to thermodynamic and atmospheric dynamics changes (Beniston, 2005; Houze, 2012; Insel et al., 2010; Poulsen et al., 2010). Our results indicate a systematic increase in precipitation amount in response to surface uplift due to orographic airlifting and associated cloud formation and condensation. For instance, the high elevation scenarios on the West-Central Alps focus precipitation on the western flank of the Alps and show a decreasing trend towards the Eastern Alps. Moreover, our results also show a threshold of the magnitude of elevation change (i.e., ca. 200% of modern topography increase) that triggers significant regional changes across Europe. More specifically, our W2 experiments of scenario 1 show significant changes (drier conditions) across Northern Europe and an extension of a rain shadow region on the eastern flanks of the Alps, whereas changes in the W1 experiments are comparatively mild. These spatial patterns are expected since the rainout on the western flanks of the Alps extracts moisture from the vapor masses on the windward slopes and leads to drier air masses crossing to the northern and eastern flanks of the Alps. The Alps are a relatively small orogen, positioned parallel to moisture transport, which should lead to more moist air masses spillover and flow around the Alps to the east and less distinct changes in precipitation amount (Sturm et al., 2010). Nevertheless, our calculations of vertical wind velocities (omega), relative humidity, and cloud cover (Fig. 10 and 11) suggest that our uplift scenarios induce enough orographic lifting to create notable updrafts that lead to the formation of thick clouds in the



troposphere (Houze, 2012). The higher along-strike terrain created by bulk surface uplift in the W2E2 experiment even results in a clearly defined ~500 km wavelength pattern of interchanging moisture uplift and subsidence, which corresponds to high and low cloud cover and relative humidity (Fig. 11 C, F, and I). Note that the local precipitation changes induced by varying the Eastern Alps topography are very different from those induced by varying the West-Central Alps topography (e.g., Fig. 4 E vs. Fig. 4 B). In summary, due to the size and orientation of the Alpine orogen with respect to the dominant wind fields, the regional precipitation response to diachronous surface uplift is unique and highly sensitive to the altitude of the West-Central Alps in particular.

**5.3 Impacts of diachronous surface uplift on moisture source and transport**

The back-trajectory analyses for the topography experiments reveal notable changes in air mass transport distance and pathways even though the predominant moisture source remains the North Atlantic Ocean (Rozanski et al., 1982). Our simulations demonstrate that diachronous surface uplift impacts the target regions located on the southern and eastern flanks of the Alps in particular compared to western and northern localities. More specifically, air mass trajectories tend to travel a longer distance and originate from higher atmospheric levels when surface topography is raised. For example, our trajectory analysis for the city of Graz in the W2 experiments (i.e., W2E1 and W2E0) indicates that moisture originates at higher atmospheric levels over the east of the North Atlantic and travels a shorter path to its destination. On the other hand, the W1E2 and W2E2 configurations significantly shorten the moisture transport distance. This indicates that the air mass would most likely experience less rainout due to the shorter period to reach condensation, yielding slightly higher $\delta^{18}O_p$ values compared to long-distance air mass transport. Furthermore, the W2 experiments deflect moisture from the Atlantic to the Mediterranean region before redirecting it to the Southern Alps. These changes in vapor transport are not surprising since mountain barriers would force the air to rise and, depending on the strength of the flow, the cross-barrier flow would be blocked or deflected towards the regions of precipitation (Colle, 2004; Grossman and Durran, 1984). The atmospheric conditions of the moisture source region would also influence the precipitation type and amount in the target regions (Feng et al., 2013). Therefore, our results stress the importance of considering the unique impacts of different topographic configurations on the moisture source and pathways when investigating past changes in precipitation (or $\delta^{18}O_p$).

**5.4 Impacts of the diachronous surface uplift on atmospheric flow and pressure systems**

Atmospheric teleconnections control much of the synoptic-scale atmospheric variability that is also important for the climate in mountain ranges (Barnston and Livezey, 1987; Rogers, 1990; Wallace and Gutzler, 1981). These patterns influence climate over a large geographic area and affect processes such as precipitation dynamics, storm tracks, jet stream location, atmospheric waves, and temperature (Hurrell, 1995; Woollings et al., 2010). Giorgi et al. (1997) indicate that altitude plays a significant role in determining the regional climate response to large-scale patterns like the North Atlantic Oscillation (NAO). Moreover, Wallace and Gutzler (1981) suggest that the high-pressure anomalies that persisted across the Alps during the 1980s were due to the shifting of the upper-level jet stream (i.e., the north-south dipole axis associated with the polar front) to the north.



Therefore, the non-stationarity of these recurrent pressure patterns (especially in summer) (Deininger et al., 2016) requires that
any attempt to reconstruct past hydrological cycles quantitatively should be done with knowledge about the likely changes in
atmospheric pressure patterns.

Our simulation results indicate a persistent leading mode of variability that is consistent with the NAO (Fig. 12) (Hurrell, 1995;
Hurrell and Van Loon, 1997). The modern (non-simulated) NAO exists in all seasons (Craig and Allan, 2022) but is more
prominent and stable in winter. A well-developed dipole pressure gradient between the Icelandic Low and Azores High in its
positive phase induces strong westerlies and northerly storm tracks that transport air masses from the eastern Atlantic towards
Central Europe. Such a pressure system drives colder and drier conditions across Western Greenland and the Mediterranean
region and warmer and wetter conditions across Northern Europe and some portions of the Scandinavia region. During its
negative phase, the pressure gradient is reduced, which causes a decrease in the strength of the westerlies and a southward shift
of the storm tracks. This mechanism leads to more precipitation across southern Europe and the Mediterranean and colder and
drier climates across northern Europe (Barnston and Livezey, 1987; Hurrell and Van Loon, 1997). Overall, our topography
experiments suggest a northeastward shift of the positive anomaly center of the action close to Ireland and the UK when the
Eastern Alps (W1E2) are uplifted and more eastward stretch with intensification across eastern Europe when the West-Central
Alps are at a maximum elevation (e.g., W2E1). The shifts of the NAO dipole axis, intensification, and location of the center
of action in response to topography changes would alter the moisture and heat transport pathways, wind patterns, and the
intensity of storms and precipitation patterns across the North Atlantic and its surrounding continents. For instance, the
eastward shift and intensification of the positive anomalies over Central Europe in response to higher west-central Alps
topography would likely lead to the reduction of rainfall across the central and southern parts of Europe due to the northward
flow of moisture.

The pressure patterns of the second mode of variability in response to the topography changes resemble the East Atlantic
pattern (EA), as originally identified by Wallace and Gutzler (1981). The exact nature of the EA pattern is still debated. While
some studies define it as a southward shift of the NAO, showing the north-south dipole pressure gradient with centers of action
across the North Atlantic from east to west (Bastos et al., 2016; Chafik et al., 2017), others define it as a well-defined monopole
pressure anomaly close to Ireland (Comas-Bru and McDermott, 2014; Josey and Marsh, 2005; Moore et al., 2013; Zubiate et
al., 2017). Our simulated pattern matches the latter description best. Such a pattern would lead to wetter conditions across
Eastern Europe and a drier climate over western Europe (Barnston and Livezey, 1987). The E0 configurations (Fig. 13 B and
D) result in the most significant changes to the second mode of variability, which highlights the potentially significant impact
of delayed Eastern Alps uplift on atmospheric pressure patterns and associated changes in precipitation across Europe. An
example of such influence on precipitation is the extensive drying across Eastern Europe in response to the W2E0 topography
configuration (Fig. 4 D)

The second and third modes of variability show some similarity to the Scandinavian pattern (SCAN) and the East
Atlantic/Western Russia pattern (EA/WR) as described in other studies (Barnston and Livezey, 1987; Comas-Bru and





McDermott, 2014; Ionita, 2014; Lim, 2015). The simulated patterns do not show any clear trends or large and systematic changes in response to the different topographic forcings.

In summary, our results suggest that different topographic configurations, including those describing diachronous surface
uplift, can induce significant changes to synoptic-scale atmospheric pressure systems in the Northern Hemisphere. Quantifying the impacts of these changes on regional climate would require an in-depth investigation of atmospheric dynamics and scale interactions in the region and is beyond the scope of this study. Our results highlight plausible changes in atmospheric pressure patterns that would significantly affect the spatial distribution of precipitation across Europe and change the source region and pathways for moisture carried onto the continent.

**5.5 Impact of regional climate changes on $\delta^{18}O_p$ and isotopic lapse rates**

The sensitivity experiments show that the topographic configurations describing diachronous surface uplift (scenario 1) affect $\delta^{18}O_p$ values across the Alps. Specifically, the W2 experiments predict an additional decrease in $\delta^{18}O_p$ values of ~ 8 ‰ on the western flanks of the Alps and a less significant change of ~ 1 to 2 ‰ in the adjacent low-elevation regions around the Alps (Fig. 2). Moreover, the simulations predict an expansion of the area of low $\delta^{18}O_p$ values (adjacent to the western side of the
orogen) when high elevation (W2) is assumed for the West-Central Alps (or the complete orogen). This is not surprising since the predominant moisture transport from west to east would be blocked or deflected by the steeper topography creating an "isotopic rain shadow." In contrast, the moderate (present-day) height allows more spill-over of moisture over and around the western Alps, especially since the Alps are positioned parallel to wind trajectories (Sturm et al., 2010). The decrease of $\delta^{18}O_p$ values over the Northern Alps, on the other hand, could be a result of the cross overflow of moisture transported from the
Mediterranean Sea. However, our trajectory analysis did not indicate a plausible moisture source from the Mediterranean in summer, which might be due to the limited model resolution to capture all the relevant air mass sources. In general, the modeled differences in $\delta^{18}O_p$ values can be attributed to changes in precipitation and temperature. These can, in turn, be explained by direct altitude-related differences between our topographic scenarios, as well as by indirect effects related to changes in the wind trajectories, changes in the vertical tropospheric structure in an orogen located in a westerlies-dominated wind field, and
changes in synoptic-scale atmospheric dynamics (sections 5.1 - 5.4). These changes differ significantly between the topographic configurations and ultimately result in different ILRs. More specifically, the annual ILRs are more negative (steeper) than the summer ILRs. These changes in ILRs are likely due to two effects: (1) The increase of isotopic fractionation with decreasing temperature (Dansgaard, 1964; Gat, 1996) affects the winter (and thus the annual) ILR, and (2) evaporative recycling of warmer surface waters in the summer leads to a different isotopic composition of the continental moisture source
(Risi et al., 2013).

Overall, this study's experiments outline that diachronous surface uplift (i.e., the west-to-east surface uplift propagation) across the Alps would have produced distinct spatial profiles of $\delta^{18}O_p$ due to both direct (altitudinal) and indirect climatic effects. If the magnitude of change in $\delta^{18}O_p$ values presented here for different topographic scenarios is preserved in geologic archives such as paleosol carbonate nodules or hydrous silicates, then the stable isotope record of these changes holds the potential to





reconstruct the hypothesized diachronous surface uplift history of the Alps. Furthermore, our results suggest that the ILR, which for the lack of tracking it through time, is often assumed to be constant in stable isotope paleoaltimetry studies, may change across the Alps depending on the specific topographic configuration. For instance, the W2E1 topographic configuration, which best matches the paleoelevation reconstruction in the middle Miocene by Krsnik et al., 2021 would correspond to an increase of ~ 0.5 ‰/km and ~ 1 ‰/km across the western and northern flanks compared to present-day

topography.

### 5.6 Model limitations and implications

The modeled present-day climate conditions are in good agreement with observational data and the expected climate patterns across Europe, as also indicated by other studies (Langebroek et al., 2011; Werner et al., 2011). The slight deviations across the Alps are likely a result of the model's underrepresentation of the subgrid topographic features (e.g., Alps ridges and slopes).

Nevertheless, the reader should carefully consider further limitations of the model and this study. ECHAM5-wiso, like other GCMs, has several deficiencies in parameterization schemes and simplifications of the underlying physics (Roeckner et al., 2003; Werner et al., 2011). Most importantly for this study, the model simplifies complex topography by smoothing high-elevation peaks, which leads to an underestimation of $\delta^{18}O_p$ values at higher elevations. Furthermore, it uses the hydrostatic approximation, which generally results in a relatively poor representation of precipitation dynamics in mountain regions with

steep topographic gradients (e.g., Steppeler et al. 2003). On top of that, ECHAM5-wiso does not simulate the oceanic variables dynamically but uses prescribed sea surface temperatures (SSTs) and sea ice concentrations (SIC) from a coupled ocean-atmosphere GCM which complicates our EOF analysis to construct the atmospheric teleconnections. The model underestimates summer precipitation across the European Alps due to the parameterization of convective processes that contribute to summer rainfall (Langbroeck et al., 2011). Moreover, ECHAM5-wiso has a simple land-surface scheme that does

not allow for proper consideration of the isotopic fractionation of surface waters (Werner et al., 2011). Since water vapor from evaporative recycling of surface waters and evapotranspiration influence the $\delta^{18}O_p$ values across Europe (Rozanski et al., 1982), the reader is advised to consider this limitation when applying our model results. Furthermore, we note that the trajectory analyses of this study track air masses by disregarding their moisture content. Therefore, changes in air mass trajectories do not inherently lead to significant changes in $\delta^{18}O_p$ if, for example, all changes in atmospheric transport only affect air masses

that are moisture depleted. The reader is also made aware that this study uses fixed pre-industrial paleoenvironmental boundary conditions for the GCM topographic sensitivity experiments to isolate the topography-related $\delta^{18}O_p$ signal of simplified diachronous surface uplift scenarios for the Alps. The reader is advised that these conditions do not represent the realistic global paleoclimate condition of the time of the Cenozoic major surface uplift, which is why we refer to our results as a sensitivity analysis of potential signals. However, despite these limitations, the results of this sensitivity analysis indicate good

potential for detecting and reconstructing a diachronous elevation history in the Alps. Given this, future time-intensive efforts such as stable isotope-based paleoaltimetry data collection and more paleogeographically realistic GCM modeling studies are considered likely to be worthy endeavors.





### 5.7 Implications for paleoaltimetry reconstructions and hypothesis evaluation

Stable isotope paleoaltimetry exploits the systematic relationship between $\delta^{18}O_p$ and elevation to infer past elevation across
orogens (e.g., Chamberlain et al., 1999; Kohn and Dettman, 2007; Mulch, 2016; Quade et al., 2007; Rowley and Garzione, 2007; Sharp et al., 2005). The present-day $\delta^{18}O_p$ lapse rate in the Central Alps today is ~ 0.2 ‰/100m (Campani et al., 2012). The present study estimates a similar range of $\delta^{18}O_p$ lapse rates across the western and northern flanks of the Alps for present-day topography conditions (-0.23 ‰/100m and -0.22 ‰/100m, respectively) and a higher value (-0.31 ‰/100m) for the southern flanks (Fig. 6). The W2E1 scenario is the closest to a plausible, albeit very simplified, scenario for the Miocene since
the Alps are suggested to have reached their maximum peaks during that period. The simulation for this topography scenario estimates $\delta^{18}O_p$ lapse rates of -0.28 ‰/100m, 0.34 ‰/100m, and -0.29 ‰/100m for west, north, and south transects, respectively (Fig. 7). These results differ from our predictions of lapse rates produced by bulk surface uplift experiments (i.e., W2E2). In other words, the diachronous surface uplift of the West-Central and Eastern Alps (scenario 1) creates distinct isotopic patterns that differ from those produced by the control simulation (CTL) or the bulk surface uplift (scenario 2)
experiments. We can therefore accept our hypotheses that 1) different topographic configurations for Eastern and West-Central Alps result in regional climates and spatial distributions of $\delta^{18}O_p$ that are significantly different from those of today and that 2) different topographic configurations for Eastern and West-Central Alps result in regional climate and spatial distributions of $\delta^{18}O_p$ that are significantly different from those produced by scenarios of bulk surface uplift of the whole Alps. Therefore, if the signals produced by diachronous surface uplift are preserved in the geological record, diachronous surface uplift should
be reflected in the associated $\delta^{18}O_p$ values. This suggests that hypothesized west-to-east surface uplift propagation could be reconstructed with stable isotope paleoaltimetry that takes advantage of these archives.

The topographic changes produce less pronounced $\delta^{18}O_p$ values changes at low-elevation sites adjacent to the Alps, which is consistent with findings from experiments presented by Botsyun et al. (2020). The affected area of $^{18}O$-depletion expands geographically when high elevation is assumed on the western flanks of the Alps. Since the δ-δ paleoaltimetry approach is
based on the premise that low-elevation sites record background climate change unrelated to topographic changes, our results stress the importance of sampling low-elevation regions also at some distance from the orogen, especially for times when the orogen had likely reached significant elevation. However, sampling at low elevation near the orogenic front may underestimate rather than overestimate the past elevation (because of low elevation $\delta^{18}O_p$ values that are lower than far-field sampling sites unaffected by nearby Alpine topography). Therefore, we recommend that this study's $\delta^{18}O_p$ maps are consulted when devising
a sampling strategy to ensure that the target low-elevation sampling location lies outside the region, in which topographic changes significantly impact $\delta^{18}O_p$ values. Paleoclimate modeling with realistic paleoenvironmental conditions may be an important support for future paleoaltimetry studies. Due to the demonstrated link between topographic configurations and the atmospheric teleconnection patterns governing European climate, estimates of past climate and $\delta^{18}O_p$ lapse rates should also consider the hypothesized uplift scenarios as part of the paleogeographic and paleoelevation boundary conditions for climate
models (e.g., Zhang et al., 2015).


## 6 Conclusion

This study employs a model-based sensitivity analysis to investigate the climatic and $\delta^{18}O_p$ response to diachronous surface uplift across the Alps. The simulated results indicate that diachronous surface uplift would produce patterns of climate, $\delta^{18}O_p$

values, and isotopic lapse rates that are distinctly different from those of today and those produced by bulk surface uplift scenarios. Therefore, we can accept the hypotheses that the diachronous surface uplift of the West-Central and Eastern Alps would result in distinct regional climates and meteoric $\delta^{18}O_p$ patterns that differ from (1) present-day conditions and (2) conditions produced when the whole Alps are uplifted. If this signal is not lost during the formation of geological archives like paleosol carbonates, these archives can be used in a stable isotope paleoaltimetry approach to test the hypothesis of eastward

propagation of surface uplift in the Alps. Note that this study only quantifies the topographic signal while keeping paleoenvironmental conditions constant. Further experiments are needed to investigate the synergistic effects of combined topographic and paleoenvironmental changes and move towards plausible reconstructions of Alps topography and paleoclimate of specific times in the past. Furthermore, the next logical step to close the gap between the predicted meteoric $\delta^{18}O$ response and isotopic ratios extracted from archives is to employ proxy system models to investigate the signal

transformation that takes place between these steps. This would allow for a more accurate back-transformation that can ultimately refine paleoelevation estimates for the Alps.

### Code availability

The ECHAM5 model is available under the MPI-M Software License Agreement (https://mpimet.mpg.de/fileadmin/projekte/ICON-ESM/mpi-m_sla_201202.pdf), and the isotope tracking implementation part

(ECHAM5-wiso) is available upon request from the Alfred Wegner Institute (AWI), Germany (https://gitlab.awi.de/mwerner/mpi-esm-wiso). The LAGRANTO model used for the back-trajectory analysis can be downloaded from www.lagranto.ethz.ch. The scripts used for postprocessing, data analysis, and visualization are based on a Python package (pyClimat) available at https://doi.org/10.5281/zenodo.7143044.

### Data availability


The processed model output variables (i.e., near-surface temperature, precipitation-weighted $\delta^{18}O_p$, precipitation amount, elevation, near-surface meridional, and zonal wind velocities single level variables, and vertical wind velocity, cloud cover,

and relative humidity on pressure levels) used in this study for the different topographic scenarios are available in NetCDF format at https://doi.org/10.5281/zenodo.7143044.

**Author contribution**

DB: Conceptualization, model simulation, data analysis, visualization, and writing of the original manuscript; SGM, TAE: conceptualization, supervision, funding acquisition, manuscript review and editing; AM, MJMM, KM: funding acquisition and manuscript review and editing; and AB, SB: manuscript review and editing. All authors contribute to the discussion of the results.

**Competing interests**

The authors declare that they have no conflict of interest.

**Acknowledgment**

This research was supported by German Science Foundation (DFG) priority research program *Mountain Building Processes in Four Dimensions* (4D-MB; SPP 2017) grants EH 329/19-1 and EH329/23-1 (to T.A.E.) and MU4188/1-1 and MU4188/3-1 (to S.G.M.) as well as MU2845/6-1 and MU2845/7-1 (to A.M.) and Me4955/1-1 (to K.M.) and ME5579/1-1 (to M.J.M.M.). We thank X and Y for constructive reviews.

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
