# Peer review of "The effects of diachronous surface uplift of the European Alps on regional climate and the oxygen isotopic composition of precipitation"

_Earth System Dynamics, 2022_

## Referee Comment (RC1)

Review by J.E. Saylor

**Summary**

The authors present a sensitivity study of the effects of diachronous elevation changes in the Alps on regional climate and, specifically, the stable isotopic composition of precipitation. The sensitivity study is conducted using the ECHAM5-wiso isotope-enabled General Circulation Model. They explore a range of end-member models, ranging from no Alpine topography, to double modern elevations in the western and eastern Alps individually or uplift of the full Alps to twice their modern elevation. The authors conclude that there are significant changes in isotopic lapse rates (in addition to changes in isotopic ratios as a result of topographic change but not due to changes in lapse rate). The authors also identify changes in temperature, precipitation amounts, and atmospheric circulation related to uplift. The authors conclude that uplift of the western Als has a greater effect than uplift of the eastern Alps and that diachronous uplift can be assessed from the geologic record, given appropriate archives.

**Recommendation**

There are significant unexplored limitations to the dataset that need to be addressed before the manuscript can be published. These limitations may undermine, or at least put caveats on, the authors' conclusions that should be incorporated into the Abstract, Discussion, and Conclusions. I recommend that the authors address the considerations below in a major revision prior to publication.

**General comments**

The authors need to calculate and present uncertainties associated with their lapse rates. It is difficult or impossible to determine if the difference in lapse rates is significant without some estimate of the uncertainty associated with the values.

The authors conclude that changing the topographic configuration changes the d18O values across the region. This goes without saying and is the basis for paleoaltimetry. The question is whether the d18O values change more or less than is expected given a certain amount of topographic rise. The authors have not demonstrated that that is the case based on my evaluation of Figure 5, 6 or 7.

There are places where the authors apparently favor non-uniformitarian interpretations based on data that are equivocal (see comment on Line 519 below). It seems that the most conservative interpretation should be favored where possible and unless the data require alternative interpretations.

**Detailed comments**

Lines 15, 117: Delete the "e.g.,"

Line 22: What is "significant"? This should be presented in terms of absolute lapse rates and their uncertainties.

Line 25: Obviously the absolute values change and that that change will vary if part or all of the orogen is uplifted, the question is whether the underlying lapse rates in isotopic ratios or temperatures change.

Line 70: Rephrase as "remains an open question."

Line 103: It would be useful to have a succinct statement of the modern elevations and the basis for selecting the elevations selected for the experiments. The latter is disseminated through this section, but it would be useful to have it stated concretely and in one location.

Line 108: What is deemed unlikely about the topographic development? It seems like this sentence needs an additional clarifying phrase.

Line 108: Should this be, "between 200 and 100 km"?

Line 108: What is meant by "post crustal shortening"?

Line 310: What about W0E0?

Line 314: I recommend rephrasing as, "The topographic scenarios predict significant localized cooling or warming where the topography is raised or lowered, respectively.

Line 315: Are these adiabatic or non-adiabatic cooling or warming?

Line 330: Consider annotating the legend with text to guide the reader, such as "Warmer than control," or "Cooler than control." Consider something similar for figure 2 and 4.

Line 370: It is quite difficult to correlate between the legends and the curves. I think this is in part because the topography is semi-transparent and so the colors are washed out. As the colors selected in the legend are somewhat similar, it makes it hard to tell the difference when they are semi-transparent. I recommend making all lines 100% opaque and perhaps labelling individual curves to aid visual correlation.

Line 370: Include these cross-sections or swaths on figure 2.

Line 370: What about W2E2? How do the lapse rates change for that scenario?

Line 380 and figure 6: What are the uncertainties in these lapse rates? I suspect that a lapse rate of -2.08 per mil per km is virtually indistinguishable from -1.83 per mil per km. Ditto for -3.11 per mil per km and -2.96 per mil per km.

Line 390: I'm not sure that linear lapse rates are appropriate here. From Figure 5 it looks like there are very different lapse rates between 0–500 m and >500 m. A back-of-the-envelope calculation of lapse rate for the W2E1 and control scenario yields similar lapse rates above 500 m. CTL: (-7.5 - -4.5)/(1.25-.5) = -4 per mil / km; W2W1: (-14 - -4.5)/(3-.5) = -3.8 per mil / km

Line 519: This 2 C is well within the range of lapse rates cited above (4.1–5.9 C/km). Without further examination of the data, it seems like an over-interpretation to invoke non-adiabatic processes here. In other words, the non-adiabatic processes must be demonstrated and not simply invoked. No such demonstration is offered here.

Line 520: Yes, plausible and the simplest explanation prima facie.

Line 523: What remainder of the signal? I am not sure what is being referred to here.

Line 525: "All of the signal can be explained via adiabatic processes. Other processes appear to be insignificant." The "small contribution" has not been demonstrated and should not be invoked without caveats.

Line 554: I can see the higher atmospheric origin (maybe…) but the paths are not convincingly longer than the CTL experiment. Also, specify that you are referring to the topography of the western Alps (obviously W2E0 has topography both raised and lowered).

Line 555: An origin at higher atmospheric levels when the topography is lowered seems to be the opposite of what was stated in the previous sentence. Reference the relevant figure.

Line 557: They also lower the elevation of the vapor source if I am interpreting the figures correctly.

Line 617: To what is the 8 per mil additional? Perhaps replace "an additional" with "a"?

Line 617: But are these differences in d18O values unexpected, or are they what would be predicted based on increasing topography without significantly changing the lapse rates? It looks like the latter based on my evaluation of Figure 5, for example.

Line 630: Where is this difference in summer versus annual lapse rates shown? The text needs to refer to specific figures and panels to support statements like this.

Line 640: Again, is the change significant? What are the uncertainties? Are the uncertainties greater than the calculated change in lapse rates?

Line 645: Where is this shown?

Line 682–684: Whether they differ or not depends on the uncertainties associated with these measurements.

---

## Author Comment (AC2)

Response to Reviewer 1 (J. E Saylor)
Reviewer's comments are repeated in black. Authors' replies are highlighted in blue font and the revised texts in the manuscript are in quotation marks with blue italics font.

We thank J. E Saylor for the constructive comments and time for highlighting parts of the manuscript that require changes and clarification.

The authors present a sensitivity study of the effects of diachronous elevation changes in the Alps on regional climate and, specifically, the stable isotopic composition of precipitation. The sensitivity study is conducted using the ECHAM5-wiso isotope-enabled General Circulation Model. They explore a range of end-member models, ranging from no Alpine topography, to double modern elevations in the western and eastern Alps individually or uplift of the full Alps to twice their modern elevation. The authors conclude that there are significant changes in isotopic lapse rates (in addition to changes in isotopic ratios as a result of topographic change but not due to changes in lapse rate). The authors also identify changes in temperature, precipitation amounts, and atmospheric circulation related to uplift. The authors conclude that uplift of the western Als has a greater effect than uplift of the eastern Alps and that diachronous uplift can be assessed from the geologic record, given appropriate archives.

The reviewer highlighted in the summary that the manuscript concludes with significant changes in the isotopic lapse rates in response to the topographic changes but not due to the changes in "lapse rate". In order to avoid possible misunderstandings, we would like to point out the following: If we refer to the "lapse rate" as the expected changes in isotope ratios due to the topographic increase or decrease, then the manuscript highlights that the simulated changes are driven by the combined influence of the linear feedback (such as altitude effect) and other non-linear responses such as redistribution of precipitation, atmospheric circulation changes, vapor transport, non-adiabatic temperature effects, etc. (For similar findings, see Ehlers and Poulsen, 2009; Insel et al., 2010; Feng et al., 2013; Lee and Fung, 2008; Risi et al., 2013; Botysun and Ehlers, 2021). On the other hand, if we refer to the "lapse rate" as temperature lapse rate, then our analysis indicates a significant decrease (increase) in localized near-surface temperature in response to an increase (decrease) in elevation. Considering the expected changes in temperature due to the elevation changes, we attribute the highly localized changes primarily to the adiabatic temperature lapse rate changes with only minor contributions from the non-adiabatic temperature changes. We demonstrate the additional changes aside from the direct lapse rate response for temperature and $\delta^{18}O_P$ in our replies to the detailed comments section.

Recommendation
There are significant unexplored limitations to the dataset that need to be addressed before the manuscript can be published. These limitations may undermine, or at least put caveats on, the authors' conclusions that should be incorporated into the Abstract, Discussion, and Conclusions. I recommend that the authors address the considerations below in a major revision prior to publication.

We thank the reviewer for suggesting that we incorporate some uncertainties associated with our analysis. We describe in our responses below how exactly we addressed the reviewer's concerns about the study's limitations.

General comments
The authors need to calculate and present uncertainties associated with their lapse rates. It is difficult or impossible to determine if the difference in lapse rates is significant without some estimate of the uncertainty associated with the values.

We agree with the reviewer's assessment and updated the manuscript accordingly. We have added uncertainties associated with the lapse rates in all the related figures (Figures 6 and 7).

The authors conclude that changing the topographic configuration changes the d18O values across the region. This goes without saying and is the basis for paleoaltimetry.

We believe there has been a small misunderstanding due to a lack of clarity in the text. Even though the manuscript also addresses the question of how much (and where) different scenarios of differentiated west-to-east surface uplift would affect $\delta^{18}O_P$, one of our goals is to help determine whether the estimated $\delta^{18}O_P$ changes in response to the different configuration of the Alps would be significant enough to be reflected in the paleoaltimetry or paleoclimate records. In that case, paleoaltimetry can be regarded as a valid tool to reconstruct paleoelevations and help understand the geodynamic evolution of the Alps, which is still debated. We have clarified this goal (lines 84-85).

"The simulated $\delta^{18}O_P$ signal can help determine if the changes are significant enough to be reflected in paleoaltimetry records, which would ultimately help to reconstruct the geodynamic evolution of the Alps."

Furthermore, the expected $\delta^{18}O_P$ changes we calculate include non-adiabatic changes that would be missed in classic methods used for paleoaltimetry. These changes constitute an important part of our conclusion and provide significant added value to well-established paleoaltimetry methods. Please see the responses below for more details on this.

The question is whether the d18O values change more or less than is expected given a certain amount of topographic rise. The authors have not demonstrated that that is the case based on my evaluation of Figure 5, 6 or 7.

This is a very valid and important concern. We addressed this as follows: since the expected $\delta^{18}O_P$ in response to certain topographic rises can be determined with a specific lapse rate, we adopt the present-day isotopic lapse rate of 2.0 ‰/km (as estimated from the isotopic observation across the Central Alps by Campani et al. (2012)). This modern isotopic lapse rate has been used to infer the past elevation of the Central Alps in the Middle Miocene (e.g., Krsnik et al. 2021). We calculate the difference between the simulated annual $\delta^{18}O_P$ changes and the expected changes in $\delta^{18}O_P$ based on the topographic changes using this particular lapse rate. The results show up to 4 ‰ differences in $\delta^{18}O_P$ values across the region of modified topography, and differences of ca. 2 ‰ for adjacent low-elevation areas. This indicates that part of the $\delta^{18}O_P$ changes in the model is due to effects that are not (directly) related to altitude changes. These additional changes in $\delta^{18}O_P$ would have been missed using a simple lapse rate to calculate

the changes based on the topographic rise. We added these results demonstrated for some of the topographic configurations to the supplementary material (Figure S16).

[Figure]

Fig. R1: Annual long-term difference between the simulated and expected change of $\delta^{18}O_P$ values in response to elevation changes across the Alps (e.g., W2E1 (a), W1E0 (b), W2E0 (c), and W1E2 (d)). The expected changes are calculated using a modern isotopic lapse rate based on long-term isotopes in precipitation measurements (Campani et al., 2012). The differences highlight the signal that would be missed if only the fixed lapse rate were used. For instance, negative difference indicates that paleoelevation would be overestimated due to their associated shallow lapse rate.

There are places where the authors apparently favor non-uniformitarian interpretations based on data that are equivocal (see comment on Line 519 below). It seems that the most conservative interpretation should be favored where possible and unless the data require alternative interpretations.
Please see our response to line 519 below.

Detailed comments
 Lines 15, 117: Delete the "e.g.,"
This has been corrected

Line 22: What is "significant"? This should be presented in terms of absolute lapse rates and their uncertainties.

The uncertainties are now provided throughout the manuscript (Figures 6 and 7). Moreover, we modified the text (e.g., in lines 23-26) to highlight that what is significant depends on the topographic rise and configuration.

"The simulated responses to the varied topography suggest changes in the spatial patterns of $\delta^{18}O_p$, the elevation-dependent rate of change in $\delta^{18}O_p$ ("isotopic lapse rate"), near-surface temperatures, precipitation amounts, and atmospheric circulation patterns. However, the magnitude and spatial patterns of the simulated changes varied, depending on the topographic configuration and the extent of the surface uplift."

Line 25: Obviously the absolute values change and that that change will vary if part or all of the orogen is uplifted, the question is whether the underlying lapse rates in isotopic ratios or temperatures change.

Yes, changes in elevation would change the $\delta^{18}O_p$ values. However, considering the size of the Alps and its geographic location with respect to moisture transport, the underlying question is whether varying the topography of the different sections would result in different (uneven or unexpected) spatial patterns and lapse rates. For instance, does changing the elevation of the West-Central Alps affect the lapse rate of air mass transporting towards the northern transect? We have demonstrated in the lapse rate estimates (Fig. 6 and 7) that the magnitude of changes varied spatially in response to the different topographic configurations.

Line 70: Rephrase as "remains an open question."

This has been corrected.

Line 103: It would be useful to have a succinct statement of the modern elevations and the basis for selecting the elevations selected for the experiments. The latter is disseminated through this section, but it would be useful to have it stated concretely and in one location.

We thank the reviewer for this insight. Additional sentences have been added to explain further the reasons for the topographic experiments (lines 125-130).

"Therefore, to explore the plausibility of stable isotope paleoaltimetry estimates in addressing the west-to-east surface uplift scenarios, we use the different topographic configurations in sensitivity experiments to quantify the expected isotopic signal. With the present-day mean elevation of peaks of ca. 2500 m across the Alps, increasing its elevation by 200% would reflect the paleoelevation reconstructions across the West-Central Alps in the middle Miocene (Campani et al., 2012; Krsnik et al., 2021). However, due to the lack of quantitative paleoelevation estimates across the Eastern Alps, we incrementally increase the elevation across that transect to explore all the potential surface uplift magnitude back in time (see Sect. 3.2 for more details about topographic configuration)."

Line 108: What is deemed unlikely about the topographic development? It seems like this sentence needs an additional clarifying phrase. Line 108: Should this be, "between 200 and 100 km"?

It's considered unlikely since significant continental lithosphere subduction under the European plate in response to the subsequent collision of the Adriatic and European plates at ca. 35 Ma after its commencement in the late Cretaceous (Schlunegger and Kissling, 2015) resulted in post-collisional crustal shortening of 100-200 km (and presumably some thickening) (Lippitsch et al., 2003; Rosenberg et al., 2015). However, we have modified the section to focus more on paleoaltimetry estimates.

Line 108: What is meant by "post crustal shortening"?
We thank the reviewer for pointing this out. We meant "post-collisional crustal shortening and refer to it as the subsequent collision that occurred between the European and Adriatic plates in the Oligocene after its initiation in the late Cretaceous to early Tertiary period.

Line 310: What about W0E0?
The W0E0 is not shown since it was similar to the results already presented by Botsyun et al., 2020. We have added a statement about this in lines 340-343 to refer readers to Botsyun et al., 2020.

"The experiment with no Alps (W0E0) predicts an increase in $\delta^{18}O_p$ values up to 8 ‰ (not shown) and was similar to the results presented by Botsyun et al., 2020. Therefore, we do not further discuss its results and refer the reader to Botsyun et al. (2020) for more details."

Line 314: I recommend rephrasing as, "The topographic scenarios predict significant localized cooling or warming where the topography is raised or lowered, respectively.
We followed this recommendation and changed the text accordingly.

Line 315: Are these adiabatic or non-adiabatic cooling or warming?
The presented values are differences in the near-surface temperature, which are a result of both (a) adiabatic warming or cooling directly linked to elevation increase or decrease, and (b) non-adiabatic warming or cooling associated with climate change (due to changes in atmospheric circulation, radiative and/or surface heating)

Line 330: Consider annotating the legend with text to guide the reader, such as "Warmer than control," or "Cooler than control." Consider something similar for figure 2 and 4.
All the simulated difference figures (Figures 2, 3, and 4) are adjusted accordingly.

Line 370: It is quite difficult to correlate between the legends and the curves. I think this is in part because the topography is semi-transparent and so the colors are washed out. As the colors selected in the legend are somewhat similar, it makes it hard to tell the difference when they are semi-transparent. I recommend making all lines 100% opaque and perhaps labelling individual curves to aid visual correlation.
We thank the reviewer for pointing this out. We have added solid (opaque) lines to the topographic profiles and added their respective colors as the shading.

Line 370: Include these cross-sections or swaths on figure 2.
This is a very helpful request made by the reviewer. Thus, to identify better the cross-sections used for the profiles, we have added the swaths on the original topography used for modification

on the same Figure 5. This would help the reader to get all the information related to the isotopic profiles visually.

Line 370: What about W2E2? How do the lapse rates change for that scenario?
The profiles were presented in the supplementary for the W2E2 and W0E0 (in Figure S8) since we mainly focus on the stepwise surface uplift responses.

Line 380 and figure 6: What are the uncertainties in these lapse rates? I suspect that a lapse rate of -2.08 per mil per km is virtually indistinguishable from -1.83 per mil per km. Ditto for - 3.11 per mil per km and -2.96 per mil per km.
Even though the correlation coefficients ($R^2$), which quantify the accuracy of the regression fitting used to determine the gradients (or lapse rate) were presented, we additionally estimate the slope error and further show the associated uncertainties of the regression fit with the threshold of 95% confidence interval with the bootstrapping technique. Overall, the isotopic lapse rate uncertainties range from 0.09 to 0.24 ‰/km, which are insignificant compared to the magnitude of the lapse rate estimated (ca. 1.5-3.4 ‰/km) (see line 400). We have added the uncertainties associated with the estimates in Figures 6 and 7. Moreover, we also mentioned that changes in the lapse rates depend on the topographic rise, configuration, and the transect used for the lapse rate estimation (see lines 690-695). For example, the surface uplift of the West-Central Alps has a higher spatial impact due to the predominant moisture trajectory path from the North Atlantic Ocean towards the Alps. In that case, estimates across the northern transect would experience changes in lapse rate due to the redistribution of precipitation caused by the orographic barrier. For example, the isotopic lapse rate difference between the W2E1 and CTL across the north transect is estimated to be 1.19 (±0.1) ‰/km (Fig. 7B). Such an estimate of difference is significant and can lead to paleoelevation uncertainties of about 0.5-1 km depending on the lapse rate used (see L915).

"However, compared to the difference in lapse rate of 0.2±0.1 ‰/km estimated across the southern flanks indicate that the impact on the isotopic lapse rate changes depends on the topographic rise and configuration. In this scenario, the northern transect lapse rates estimate a higher magnitude of change of lapse rate since the higher topography established across the west-central Alps redistributes precipitation due to the orographic barrier to the moisture trajectories paths from the North Atlantic Ocean, which cause dryness toward the north (Fig. 4 B)."

Line 390: I'm not sure that linear lapse rates are appropriate here. From Figure 5 it looks like there are very different lapse rates between 0–500 m and >500 m. A back-of-the-envelope calculation of lapse rate for the W2E1 and control scenario yields similar lapse rates above 500 m. CTL: (-7.5 - -4.5)/(1.25-.5) = -4 per mil / km; W2W1: (-14 - -4.5)/(3-.5) = -3.8 per mil / km
We rely on empirical isotopic lapse rates using least square regression. Using a specific profile would not reflect the wide-area lapse rate. Besides, different points along the profile would also lead to different estimates. We refer the reviewer to previous studies that have used a similar technique (e.g., Rowley, 2007; Poage and Chamberlain, 2001; Feng and Poulsen, 2016). We deem these techniques more suitable for a regional study such as ours.

Line 519: This 2 C is well within the range of lapse rates cited above (4.1–5.9 C/km). Without further examination of the data, it seems like an over-interpretation to invoke non-adiabatic processes here. In other words, the non-adiabatic processes must be demonstrated and not simply invoked. No such demonstration is offered here.

We thank the reviewer for pointing this out. To quantitatively support our discussion further, we first estimate the expected temperature change attributed to adiabatic cooling or warming using a lapse rate of 5.6 °C/km, as estimated for the Alps by previous studies (e.g., Kirchner et al., 2009; Rolland, 2003) and our own simulation. We then subtract the adiabatic cooling or warming from the total temperature change. For example, the W2E1 experiment estimates non-adiabatic cooling of up to -2°C across the Alps and its adjacent regions. We have added this work to the text (see lines 545-555) and added a supporting figure in the supplementary material (Fig. S17).

"To validate the potential non-adiabatic cooling or warming due to the regional climate change, we first estimate adiabatic temperature change using a lapse rate of 5.6 °C/km, which is within the range reported values in previous studies using the CTL experiment (Kirchner et al., 2009; Rolland, 2003). We then subtract this adiabatic temperature change from the total temperature changes to determine the contribution of non-adiabatic processes to the total changes. Overall, we notice non-adiabatic cooling of up to -2 °C across the Alps in response to topographic rise, and ca. -1 °C in the adjacent remote regions (Fig. S17). Previous studies on major mountain ranges such as the Andean Plateau and North American Cordilleran also suggest additional non-adiabatic temperature changes in response to surface uplift (Ehlers and Poulsen, 2009; Feng and Poulsen, 2016). The non-adiabatic warming or cooling in response to the different topographic scenarios may be as a result of the associated regional climatic changes, such as changes in tropospheric dynamics, local atmospheric humidity, and atmospheric circulation patterns (sections 5.3 and 5.4)."

[Figure]

Fig. R2: Annual changes of non-adiabatic near-surface temperature between the topographic scenarios (e.g., W2E1 (a), W1E0 (b), W2E0 (c), and W1E2 (d)) and CTL. The adiabatic lapse rate of 5.6°C/km was used to calculate the expected adiabatic cooling or warming due to the elevation changes and then subtracted from the total temperature changes.

Line 520: Yes, plausible and the simplest explanation prima facie.
This has been addressed in the previous comments (response to line 519 comment).

Line 523: What remainder of the signal? I am not sure what is being referred to here.
We referred the remainder of the signal to the temperature difference between the total temperature changes and the temperature change due to the adiabatic lapse rate. We have modified the text (line 554).

Line 525: "All of the signal can be explained via adiabatic processes. Other processes appear to be insignificant." The "small contribution" has not been demonstrated and should not be invoked without caveats.
This has been clarified in the previous comment (response to line 519 comment).

Line 554: I can see the higher atmospheric origin (maybe…) but the paths are not convincingly longer than the CTL experiment. Also, specify that you are referring to the topography of the western Alps (obviously W2E0 has topography both raised and lowered).
We thank the reviewer for highlighting this, and we agree. We have modified the sentence and added the reference figures to support the discussion.

Line 555: An origin at higher atmospheric levels when the topography is lowered seems to be the opposite of what was stated in the previous sentence. Reference the relevant figure.
Agreed. This has been corrected.

Line 557: They also lower the elevation of the vapor source if I am interpreting the figures correctly.
Agreed. This has been corrected.

Line 617: To what is the 8 per mil additional? Perhaps replace "an additional" with "a"?
We deleted the "additional" to resolve the complication of the sentence.

 Line 617: But are these differences in d18O values unexpected, or are they what would be predicted based on increasing topography without significantly changing the lapse rates? It looks like the latter based on my evaluation of Figure 5, for example.
This has been addressed in the previous comments  (response to line 519 comment).

Line 630: Where is this difference in summer versus annual lapse rates shown? The text needs to refer to specific figures and panels to support statements like this.
These were not shown in the main manuscript. Thanks for pointing this out. We also agree that these need to be referenced. The appropriate figures presented in the supplementary (Fig. S9 and S10) are added to the sentence.

Line 640: Again, is the change significant? What are the uncertainties? Are the uncertainties greater than the calculated change in lapse rates?
We thank the reviewer for raising such a valid point. We have shown in our response to line 380 that the estimated lapse rate uncertainties are not significant compared to the lapse rate magnitudes.

Line 645: Where is this shown?
The figures of the lapse rate estimates are now referenced in the updated sentence.

Line 682–684: Whether they differ or not depends on the uncertainties associated with these measurements
We have addressed this in the previous comment (line 380) and highlighted that the uncertainties are not significant.

REFERENCES

Botsyun, S. and Ehlers, T. A.: How Can Climate Models Be Used in Paleoelevation Reconstructions?, Frontiers in Earth Science, 9, https://doi.org/10.3389/feart.2021.624542, 2021.

Campani, M., Mulch, A., Kempf, O., Schlunegger, F., and Mancktelow, N.: Miocene paleotopography of the Central Alps, Earth and Planetary Science Letters, 337–338, 174–185, https://doi.org/10.1016/j.epsl.2012.05.017, 2012.

Ehlers, T. A. and Poulsen, C. J.: Influence of Andean uplift on climate and paleoaltimetry estimates, Earth and Planetary Science Letters, 281, 238–248, https://doi.org/10.1016/j.epsl.2009.02.026, 2009.

Feng, R. and Poulsen, C. J.: Refinement of Eocene lapse rates, fossil-leaf altimetry, and North American Cordilleran surface elevation estimates, Earth and Planetary Science Letters, 436, 130–141, https://doi.org/10.1016/j.epsl.2015.12.022, 2016.

Feng, R., Poulsen, C. J., Werner, M., Chamberlain, C. P., Mix, H. T., and Mulch, A.: Early Cenozoic evolution of topography, climate, and stable isotopes in precipitation in the North American Cordillera, American Journal of Science, 313, 613–648, https://doi.org/10.2475/07.2013.01, 2013.

Insel, N., Poulsen, C. J., Ehlers, T. A., and Sturm, C.: Response of meteoric δ18O to surface uplift — Implications for Cenozoic Andean Plateau growth, Earth and Planetary Science Letters, 317–318, 262–272, https://doi.org/10.1016/j.epsl.2011.11.039, 2012.

Krsnik, E., Methner, K., Campani, M., Botsyun, S., Mutz, S. G., Ehlers, T. A., Kempf, O., Fiebig, J., Schlunegger, F., and Mulch, A.: Miocene high elevation in the Central Alps, Solid Earth, 12, 2615–2631, https://doi.org/10.5194/se-12-2615-2021, 2021.

Lee, J.-E. and Fung, I.: "Amount effect" of water isotopes and quantitative analysis of post-condensation processes, Hydrol. Process., 22, 1–8, https://doi.org/10.1002/hyp.6637, 2008.

Lippitsch, R.: Upper mantle structure beneath the Alpine orogen from high-resolution teleseismic tomography, J. Geophys. Res., 108, 2376, https://doi.org/10.1029/2002JB002016, 2003.

Poage, M. A. and Chamberlain, C. P.: Empirical Relationships Between Elevation and the Stable Isotope Composition of Precipitation and Surface Waters: Considerations for Studies of Paleoelevation Change, American Journal of Science, 301, 1–15, https://doi.org/10.2475/ajs.301.1.1, 2001.

Risi, C., Noone, D., Frankenberg, C., and Worden, J.: Role of continental recycling in intraseasonal variations of continental moisture as deduced from model simulations and water vapor isotopic measurements: Continental Recycling and Water Isotopes, Water Resour. Res., 49, 4136–4156, https://doi.org/10.1002/wrcr.20312, 2013.

Rosenberg, C. L., Berger, A., Bellahsen, N., and Bousquet, R.: Relating orogen width to shortening, erosion, and exhumation during Alpine collision, Tectonics, 34, 1306–1328, https://doi.org/10.1002/2014TC003736, 2015.

Kirchner, M., Faus-Kessler, T., Jakobi, G., Levy, W., Henkelmann, B., Bernhöft, S., Kotalik, J., Zsolnay, A., Bassan, R., Belis, C., Kräuchi, N., Moche, W., Simončič, P., Uhl, M., Weiss, P., and Schramm, K.-W.: Vertical distribution of organochlorine pesticides in humus along Alpine altitudinal profiles in relation to ambiental parameters, Environmental Pollution, 157, 3238–3247, https://doi.org/10.1016/j.envpol.2009.06.011, 2009.

Rolland, C.: Spatial and Seasonal Variations of Air Temperature Lapse Rates in Alpine Regions, J. Climate, 16, 1032–1046, https://doi.org/10.1175/1520-0442(2003)016<1032:SASVOA>2.0.CO;2, 2003.

Rowley, D. B. and Garzione, C. N.: Stable Isotope-Based Paleoaltimetry, Annual Review of Earth and Planetary Sciences, 35, 463–508, https://doi.org/10.1146/annurev.earth.35.031306.140155, 2007.

Schlunegger, F. and Kissling, E.: Slab rollback orogeny in the Alps and evolution of the Swiss Molasse basin, Nat Commun, 6, 1–10, https://doi.org/10.1038/ncomms9605, 2015.

---

## Author Comment (AC3)

Response to Reviewer 2 (Anonymous)
Reviewer's comments are repeated in black. Authors' replies are highlighted in blue font and the revised text in the manuscript is in quotation marks with blue italics font.

The study by Boateng et al explored the application of water isotope tracking enabled AGCMs to simulate climate and precipitation d18O responses to different uplift scenarios of the Alps with the goal to identify potential imprints in d18O that can be used to reconstruct paleo-elevations of this orogen. The manuscript is pretty easy to follow, and the results are straightforward and interesting.

My main suggestion is that the analysis of the influences of different elevation scenarios of Alps on the modes of decadal pressure variability can be better linked to the precipitation and trajectory analysis. In geological archives, you will only see changes of the mean states of precipitation and d18O given the loose temporal resolution and signal averaging in records. If the decadal variability in pressure fields were to affect the mean state of precipitation and d18O, it needs to incur asymmetric responses of precipitation and/or moisture transport paths between positive, neutral, negative phases of those modes of variability. Alternatively, you could cut out this part of the analyses, which seemed to be independent from the rest of the analyses anyway.

We thank the reviewer for their time and the constructive comments highlighted to improve the manuscript. The reviewer raises a very valid point for their main suggestion. In fact, we agree that the changes in the atmospheric mode of variability would not be reflected in the low temporal resolution of the geologic archives. However, we performed such analysis to determine if the topography of the Alps can somewhat affect such large-scale circulations. Since European climate variability and moisture transport to the continent are mainly driven by such circulation patterns, it is worth investigating if having representative topographic scenarios would drive the seasonal and spatial distribution of $\delta^{18}O_p$ in a different way than in the present-day climate. Methner et al., 2020 suggested that the reorganization of the mid-latitude atmospheric circulation resulted in seasonal changes in the timing of carbonate formation across Central Europe in the Middle Miocene. As much as the global climate would drive such changes, such an analysis highlights the role the topography across the Alps can play in changing these circulation patterns. We have clarified this in the text in lines 648-655.

"However, we do acknowledge that changes in such a decadal mode of variability would not be reflected in such low resolution of geologic archives used for stable isotope paleoaltimetry. Nevertheless, assessing the role of topography in changing the atmospheric dynamics of such large-scale circulation patterns sheds light on the possibility of its impact on the spatial variability and distribution of $\delta^{18}O_p$. For instance, this has been highlighted by Methner et al., (2020) on the possibility of the reorganization of the mid-latitude atmospheric circulation in the Middle Miocene that led to the seasonal changes in the timing of carbonate formation across Central Europe."

More detailed suggestions:

1. Line 90, change "This" to "This event".

2.      Line 95, change "These" to "These processes"

3.      Line 98, change "slab break-off is suggested to have occurred" to "it has been suggested that slab break-off occurred under..."

4.      Line 99, "suggest" is used twice in a sentence, change to "argue"

5.      Line 100, change "considers" to "focuses on"

6.      Line 106, change "the eastern Alps initiated their" to "orographic development of eastern Alps initiated during the..."

7.      Line 110 to 111, "Reconstruction for the central Alps" change to "Reconstruction of the central Alps paleo-elevation"

8.      Line 136, remove "recent". Some of the knowledge has been around for quite a while.

All of the above points (1-8) have been corrected as suggested.

9.      Line 142, please see Feng et al., (2016, EPSL) for the "Previous studies have used GCMs to perform topographic sensitivity experiments…" as well.

We thank the reviewer for the additional references. We have added it to the list.

10.     Line 165 to 170, how was the subgrid scale topographic variability treated in those topography sensitivity experiments?

We thank the reviewer for raising such a concern. We modified the GTOPO30 Digital Elevation Model provided by the U.S. Geological Survey, which has a resolution of 30-arc seconds (ca. 1 km). Afterward, the modified high-resolution DEMs are interpolated to the ECHAM5-wiso model resolution. The associated relevant subgrid orographic variables are calculated from the higher-resolution DEM. These include variables such as the orographic standard deviation (i.e., the variability of the heights of the mountain range), anisotropy, peak elevations, valley elevations, mean slope, and orientation within a grid cell. Such related information is used for the subgrid-scale parameterization that estimates the effect of mountain-induced wave drag on the atmosphere and mountain blocking in the model (Stevens et al., 2013; Roeckner et al., 2003). We extend section 3.2 with additional sentences to clarify how the sub-grid topographic variability was treated (in lines 185-192).

"The topographic boundary conditions for the different experiments are prepared as follows: We modify the GTOPO30 Digital Elevation Model provided by the U.S Geological Survey, which has a resolution of 30-arc seconds (ca. 1 km). Afterward, the modified high resolution DEMs are interpolated to the ECHAM5-wiso model resolution. The associated subgrid orographic variables are calculated from the higher resolution DEM. These variables include orographic standard deviation (i.e., the variability of the heights of the mountain range), anisotropy, peak elevations, valley elevations, mean slope, and orientation within a grid cell. Such related information is used for the subgrid-scale parameterization that estimates the effect of mountain-induced wave drag on the atmosphere and mountain blocking in the model (Stevens et al., 2013; Roeckner et al., 2003)."

11.     Line 195 and Table 1, it is hard to visualize different scenarios of topographic evolution through time, it would be very helpful to turn this table into a topographic evolution diagram with respect to different time intervals.

We thank the reviewer for pointing this out. The topographic configuration tested through these sensitivity experiments represent plausible scenarios due to the limited quantitative paleoelevation estimates across the Alps. Moreover, the geodynamic evolution of the Alps that could shed more light on the different topographic evolution is still under debate (e.g., Schmid et al., 1996; Handy et al., 2010). Therefore, we simply conduct sensitivity experiments and aim to explore the feasibility of future paleoelevation altimetry estimates that would contribute to the understanding of the geodynamics evolution of the Alps.

12.     Line 228, expand "statistical significance of these differences" into "statistical significance of these differences against simulated interannual climate variability"

This has been corrected.

13.     Line 300, change "changes in the …" to "changes of d18Op in the…"

This has been corrected.

14.     Line 340, "<80 mm/month", how much precipitation is that?

This has been changed to the range of simulated difference which is 50-80 mm/month.

15.     Line 522 to 524, see Feng et al., (2016) for non-adiabatic temperature responses simulated in the Eocene North American Cordillera.

We thank the reviewer for the additional reference that supports our discussion. We have added it to our list.

16.     Line 625 to 626, it may worth mentioning that the configuration of paleo-Tethys would likely make a difference in whether there is "Mediterranean" moisture input or not.

We thank the reviewer for highlighting this. In the region of the Alps, the Paleo-Tethys fully subducted during the Mesozoic. The successive opening and closure of the Piemont-Ligurian and Hallstatt-Melita oceans led to the Alpine orogeny, but given that they had fully subducted by the Middle Miocene, we believe that the reviewer is referring to

Paratethys and its climatic effects. We have extended the sentence to include the potential influence of the closure of the Paratethys Sea on moisture transport (lines 665-673).

"Nevertheless, the Late Cretaceous to Paleogene closure of the Tethys Ocean, which lead to the surface uplift of the Alps might influence the transport of moisture from the Mediterranean in a past climate. Botsyun et al. (2022) simulate the global climate with middle Miocene paleoenvironment conditions while considering the Paratethys Sea extent in their land-sea mask to determine the impacts of the marine transgression on the regional climate. Their results indicate an increase in precipitation up to 400 mm/yr around the regions adjacent to the Paratethys Sea with anticyclonic circulation situated over the Mediterranean in the winter season. Note, however, a fully coupled ocean-atmosphere GCM model would be needed for a realistic assessment of the contribution of ocean circulation to the distribution of $\delta^{18}O_P$ patterns across Europe."

REFERENCES

Handy, M. R., M. Schmid, S., Bousquet, R., Kissling, E., and Bernoulli, D.: Reconciling plate-tectonic reconstructions of Alpine Tethys with the geological–geophysical record of spreading and subduction in the Alps, Earth-Science Reviews, 102, 121–158, https://doi.org/10.1016/j.earscirev.2010.06.002, 2010.

Methner, K., Campani, M., Fiebig, J., Löffler, N., Kempf, O., and Mulch, A.: Middle Miocene long-term continental temperature change in and out of pace with marine climate records, Scientific Reports, 10, 7989, 2020.

Roeckner, E., Bäuml, G., Bonaventura, L., Brokopf, R., Esch, M., Giorgetta, M., Hagemann, S., Kirchner, I., Kornblueh, L., Manzini, E., Rhodin, A., Schlese, U., Schulzweida, U., and Tompkins, A.: The atmospheric general circulation model ECHAM 5. PART I: Model description, https://doi.org/10.17617/2.995269, 2003.

Schmid, S. M., Pfiffner, O. A., Froitzheim, N., Schönborn, G., and Kissling, E.: Geophysical-geological transect and tectonic evolution of the Swiss-Italian Alps, Tectonics, 15, 1036–1064, https://doi.org/10.1029/96TC00433, 1996.

Stevens, B., Giorgetta, M., Esch, M., Mauritsen, T., Crueger, T., Rast, S., Salzmann, M., Schmidt, H., Bader, J., Block, K., Brokopf, R., Fast, I., Kinne, S., Kornblueh, L., Lohmann, U., Pincus, R., Reichler, T., and Roeckner, E.: Atmospheric component of the MPI-M Earth System Model: ECHAM6, Journal of Advances in Modeling Earth Systems, 5, 146–172, https://doi.org/10.1002/jame.20015, 2013.

---

## Referee Report (RR1)

2nd review by J.E. Saylor

**Summary**

The authors have addressed the reviewers concerns in part. However, there are still significant gaps in the data analysis and presentation as outlined below.

**Recommendation**

My primary reservations regarding the manuscript from the first round still stand. I do not recommend publication of this manuscript in its present form. I recommend that it undergo an additional round of major revisions before being reviewed a third time.

**General comments**

As in the first version, I do not think that the Abstract or Conclusions sufficiently lay out the results and associated caveats. For example, it is insufficient to indicate "changes" in the model output associated with changing topography without indicating whether those changes are within uncertainty. That the model would change with changing topography is a facile statement. Since most readers will read only the abstract, the abstract must convey both the results and some sense of whether the results are significant. For this study, I suspect that some of the results are significant but others (e.g., temperature, see comment on Lines 540, 543, and 552 below) are not.

The authors have provided uncertainties associated with their lapse rates in the form of 95% confidence intervals (presumably of the lapse rate linear fit). This is useful, but a more appropriate uncertainty would be the 95% prediction interval. Given that the lapse rate is empirically calculated, the most relevant question is whether an additional data point would be consistent with the calculated lapse rate (i.e., the prediction interval, not the confidence interval). In other words, if you want to know whether an observation is consistent with a model you want the prediction interval. As I stated previously, I still suspect that these lapse rates are indistinguishable in terms of their prediction intervals.

The treatment of temperature changes attributes very small changes in lapse rate to non-adiabatic processes. However, the small changes are within uncertainty of the adiabatic lapse rates. Therefore, although it is possible that these are non-adiabatic changes, it is at least equally likely that they are simply the result of the adiabatic lapse rate. It is impossible to distinguish these two scenarios as far as I can tell. I would favor a conservative approach in which all changes that are within uncertainty of the adiabatic lapse rate are attributed to the adiabatic lapse rate. Nevertheless, the manuscript needs to clearly state what the model results support, and what is interpretation. As it stands, these two are conflated as indicated by statements such as Line 552 in the manuscript (see also comment on Line 552 below).

**Detailed comments**

Lines 15–21: This could be condensed into 1–2 lines. For example, most of the motivation is irrelevant for the Abstract. Save the space for communicating your results and interpretation.

Line 56: The theoretical Rayleigh distillation curve is non-linear (Rowley, 2007).

Line 540: The manuscript already states that it is reasonable to attribute 80% of the temperature change to the adiabatic lapse rate. Rephrase this sentence. I suspect it is a hold-over from the first version.

Line 543: This approach absolutely needs to be dropped. If there is a potential range of temperature lapse rates, then you need to calculate the potential range of adiabatic temperature decreases based on that range in lapse rates. It is invalid to only use one value (the mean?) and then conclude that some small fraction of the total observed change must be due to non-adiabatic climate change. The best that could be argued is that the misfit _might_ be due to non-adiabatic temperature changes. But it might not be…

Line 552: No. The results suggest nothing of the sort. They suggest that all of the change can be attributed to adiabatic temperature changes, but that a small fraction _might_ be attributable to non-adiabatic changes. The signal is within the range of the noise.

Line 677: I wonder if it would be worth highlighting the fact that the only way to get d18O values more negative than ~-8 per mil is to have topography that is higher than modern.

Line 685: I don't see the 1 per mil per km. 0.5 per mil per km might be possible based on average values, but, again, the uncertainties are important.

Line 692: Without specifying what the effect of the model shortcomings are it is virtually impossible for most readers to consider these limitations in any meaningful way. I am not sure what to advise here, because I assume the effects of the model limitations are unquantified (and perhaps unquantifiable until better models are built). Nevertheless, this section reads very much like a disclaimer.

Line 720: The limitations of this section come back to the uncertainties, which in this case should certainly be the prediction intervals. The primary question is whether, given a new data point or data set, you could distinguish between the proposed models. I suspect that given the spread in the data used to calculate lapse rates the answer is no. However, the authors need to demonstrate that that is not the case.

Line 753: As I indicated in my first review, the conclusions and abstract need more detail and caveats associated with the conclusions given the fact that these are the sections that most people will read. Stating that the changes are "distinctly different" does not communicate any of the nuances of potentially overlapping lapse rates in d8O or temperature.

---

## Referee Report (RR2)

**Review by J.E. Saylor**

**Summary**
This is my third review of the manuscript by Boateng et al. The authors have addressed all of the comments raised in the previous reviews. I understand the authors' motivations for reporting confidence limits rather than prediction limits for the calculated isotopic lapse rates and the justification makes sense. Nevertheless, I appreciate that they include the confidence limits and thereby make this research as transparent as possible and also as useful as possible to future researchers.

**Recommendation**
 I recommend that this manuscript be published after the authors address the minor comments below.

**General comments**
My comments below indicate that there are a disturbing number of minor issues with the manuscript considering that it is on its third round of reviews. I recommend that the authors carefully revise the text, critically looking for minor problems and internal inconsistencies.

**Detailed comments**
Line 18: Consider replacing "for stable" with "to be detected using stable".
Lines 109 & 111: No caps for "middle" in "middle Miocene" since it is not a formal epoch or age. Do a universal search and correct throughout the manuscript.
Line 135–139: This is tricky because the atmospheric circulation interacts with topography as shown by this study. I am not sure what to recommend except to explicitly acknowledge the feedbacks between atmospheric circulation and topographic change.
Line 178: Delete, "The reader is advised that". It is condescending.
Line 183: What is the ECHAM5-wise model resolution?
Line 252: Establishing lapse rates for paleoelevation studies may not be the goal of this study, it is an obvious application of this research. Would such an application be valid? If not, why not? This doesn't affect the current study, but an explanatory statement may help future researchers use your research appropriately.
Line 259: I am confused by this statement. Are the authors stating that the lapse rates and associated uncertainties should not be used in an empirical paleoelevation study? If so, the reasoning is not clear but is quite important to how this study is used in future research. Can you expand on the reasoning for this statement or clarify this statement?
Line 294: Is the offset systematic or random? I have never found the overlays such as those presented in Figure 1B a particularly helpful way to visually evaluate the data. I would recommend a more simple plot such as a biplot of GNIP d18O vs model d18O for each of the GNIP locations. It seems like that would be much easier to evaluate the deviation of the model from empirical data.
Line 385: I see ~-6 per mil change between the CTL and W2E1 in Figure 5C at ~8 degrees E, but I am not sure that I see -8 per mill change anywhere. Can you clarify where this -8 per mil comes from?

Line 387: But these values are slightly lower in Figure 5B (~-11, ~-9, and -6 per mil). How
should the reader understand the difference between text and the figures?

Line 678: Doesn't this contradict the argument made in lines 135–139?

Line 690: Shouldn't this be a d18O value (rather than Dd18O value) based on Figures 6 and 7?

Line 773: Should this be Dd18O or d18O (see comment above on line 690)?

---

## Author Response (AR2)

Regarding: response to the reviews of our research article "**The effects of diachronous surface uplift of the European Alps on regional climate and the isotopic composition of precipitation**" by Boateng et al.

Dear Prof. Gabriele Messori,

We carefully considered and addressed all of the reviewer's remaining concerns. We revised our manuscript accordingly. The most notable changes are summarized below:

1.  We followed the reviewer's suggestion to modify the abstract and conclusions to include a more nuanced discussion of our results, interpretations and associated uncertainties.

2.  We agree with the reviewer that our "back of the envelope" calculation of adiabatic and non-adiabatic contributions to temperature changes is insufficient to say with any certainty that non-adiabatic processes play a role. We have followed the reviewer's advice to drop the approach and adjusted our discussion accordingly.

3.  We clarified the confusion around statistical uncertainties and added specific examples of significant and insignificant parts of our work to our manuscript and abstract. This will also prevent similar confusions among future readers.

4.  We have followed the reviewer's advice to calculate and include prediction intervals in our analysis. Fig. 6 and 7 were updated accordingly. However, we disagree with the reviewer that prediction intervals are more suitable and therefore still keep and emphasize confidence intervals. We explained our reasons in detail in our response.

We believe that the implemented changes will resolve all of the raised issues. We kindly ask the editor to carefully consider our detailed response and explanations, where we only partially agree with the reviewer (over point 4).

With kind regards and thanks for your contributions to the review process thus far,

Daniel Boateng and co-authors

Response to Reviewer 1 (J. E Saylor)

Reviewer's comments are repeated in black. Authors' replies are highlighted in blue font, and the revised texts in the manuscript are in quotation marks with blue italics font.

**Summary**

The authors have addressed the reviewer's concerns in part. However, there are still significant gaps in the data analysis and presentation as outlined below.

**Recommendation**

My primary reservations regarding the manuscript from the first round still stand. I do not recommend publication of this manuscript in its present form. I recommend that it undergo an additional round of major revisions before being reviewed a third time.

We thank the reviewer for their comments and suggestions regarding the parts of the manuscript that require improvement and clarification, and for their time for reviewing our manuscript the second time. We have addressed all of their comments and suggestions and have implemented the changes in the manuscript, which helped improve it further.

**General comments**

As in the first version, I do not think that the Abstract or Conclusions sufficiently lay out the results and associated caveats. For example, it is insufficient to indicate "changes" in the model output associated with changing topography without indicating whether those changes are within uncertainty. That the model would change with changing topography is a facile statement. Since most readers will read only the abstract, the abstract must convey both the results and some sense of whether the results are significant. For this study, I suspect that some of the results are significant but others (e.g., temperature, see comment on Lines 540, 543, and 552 below) are not.

We agree with the Reviewer. We have therefore modified the abstract (see lines 15-33) to highlight the key results of the study, the statistical uncertainties, as well as the finer nuances related to the isotopic lapse rates changes. We have also modified the conclusion (see lines 765-798) to present the magnitude of the changes associated with topographic alterations and their statistical significance in reference to the control experiment (see our replies to comments in lines 15-12 and 733).

The authors have provided uncertainties associated with their lapse rates in the form of 95% confidence intervals (presumably of the lapse rate linear fit). This is useful, but a more appropriate uncertainty would be the 95% prediction interval. Given that the lapse rate is empirically calculated, the most relevant question is whether an additional data point would be consistent with the calculated lapse rate (i.e., the prediction interval, not the confidence interval). In other words, if you want to know whether an observation is consistent with a model you want the prediction interval. As I stated previously, I still suspect that these lapse rates are indistinguishable in terms of their prediction intervals.

We have now further calculated the "prediction interval" on the linear regression best fit used for the lapse rate estimates, as suggested by the reviewer, and updated figures 6 and 7 to include these.

However, since the lapse rate estimates are based on simulated $\delta^{18}O_p$ values within the defined transect (fixed data points and additional data not necessary here) and are intended to be compared with similar estimates from different topographic scenarios but not as predictive models for constraining past elevation in this study. We therefore find the "confidence interval," which accounts for the mean response in the transect, more appropriate. The prediction interval of the fitted line as an additional statistical uncertainty metric does not change the reported "slope error" based on the confidence interval with the standard deviation of the slope as a point estimate. Therefore, we cannot conclude that the lapse rates are indistinguishable. We provide details in our replies to the comments lines 685 and 720.

The treatment of temperature changes attributes very small changes in lapse rate to nonadiabatic processes. However, the small changes are within uncertainty of the adiabatic lapse rates. Therefore, although it is possible that these are non-adiabatic changes, it is at least equally likely that they are simply the result of the adiabatic lapse rate. It is impossible to distinguish these two scenarios as far as I can tell. I would favor a conservative approach in which all changes that are within uncertainty of the adiabatic lapse rate are attributed to the adiabatic lapse rate. Nevertheless, the manuscript needs to clearly state what the model results support, and what is interpretation. As it stands, these two are conflated as indicated by statements such as Line 552 in the manuscript (see also comment on Line 552 below).

We agree with the Reviewer that the temperature changes due to the non-adiabatic lapse rate are very small, and that there is a possibility that all the changes can be attributed to the adiabatic lapse rate. We have modified the text to highlight this in lines 550-561. However, due to the known influence of large-scale atmospheric circulation on the region and its potential changes due to alterations in topography, we only suggest that the small temperature changes over the adjacent areas of modified topography 'might be attributed to' a non-adiabatic process. More details are provided in our response to the comment in Line 552.

**Detailed comments**
Lines 15–21: This could be condensed into 1–2 lines. For example, most of the motivation is irrelevant to the Abstract. Save the space for communicating your results and interpretation.

We have shortened the introductory lines of the abstract, and modified it to clearly indicate the manuscript's main hypothesis and the summary of the results that support the answer to the hypothesis and its implication (see lines 15-32).

*"This study presents the simulated response of regional climate and oxygen isotopic composition of precipitation ($\delta^{18}O_p$) to different along-strike topographic evolution scenarios. These simulations are conducted to determine if the previously hypothesized diachronous surface uplift in the Western and Eastern Alps would produce $\delta^{18}O_p$ signals in the geologic record that are sufficiently large and distinct for stable isotope paleoaltimetry. We present a series of topographic sensitivity experiments conducted with the water isotope tracking atmospheric General Circulation Model (GCM) ECHAM5-wiso. The topographic scenarios are created from the variation of two free parameters: (1) the elevation of the West-Central Alps and (2) the elevation of the Eastern Alps. The results indicate $\Delta \delta^{18}O_p$ values (i.e., the difference between $\delta^{18}O_p$ values at the low- and high-elevation sites) of up to -8 ‰ along the strike of the Alps for the diachronous*

*uplift scenarios, primarily due to changes in orographic precipitation and adiabatic lapse rate driven localized changes in near-surface variables. These simulated magnitudes of Δ $\delta^{18}O_p$ values suggest that the expected isotopic signal would be significant enough to be preserved and measured in geologic archives. Moreover, the simulated slight $\delta^{18}O_p$ differences of 1-2 ‰ across the low-elevation sites support the use of the δ-δ paleoaltimetry approach and highlight the importance of sampling far-field low-elevation sites to discern between the different surface uplift scenarios. The elevation-dependent rate of change in $\delta^{18}O_p$ ("isotopic lapse rate") varies depending on the topographic configuration and the extent of the surface uplift. Most of the changes are significant (e.g., -1.04 ‰/km change with slope error of ±0.09 ‰/km), while others were within the range of the statistical uncertainties (e.g., -0.15 ‰/km change with slope error of ±0.13 ‰/km). The results also highlight the plausible changes in atmospheric circulation patterns and associated changes in moisture transport pathways in response to changes in the Alps topography. These large-scale atmospheric dynamics changes can complicate the underlying assumption of stable isotope paleoaltimetry and therefore require integration with paleoclimate modeling to ensure accurate reconstruction of Alps paleoelevation."*

Line 56: The theoretical Rayleigh distillation curve is non-linear (Rowley, 2007).

We agree with the Reviewer and made no statement about the Rayleigh distillation curve being linear in our manuscript. However, we acknowledge that the use of "non-linear" climatic responses in this particular sentence may confuse the readers as though we are suggesting something different for the Rayleigh distillation curve. We therefore removed "non-linear" from the sentence, which reads now as follows (see line 56):

*"However, numerous climatic processes, such as surface recycling, aridity, vapor mixing, variability in moisture source, and precipitation dynamics, can also influence the isotopic lapse rate and thus complicate stable isotope paleoaltimetry estimates (Ehlers and Poulsen, 2009; Insel et al., 2010; Feng et al., 2013; Lee and Fung, 2008; Risi et al., 2013; Botysun and Ehlers, 2021)."*

Line 540: The manuscript already states that it is reasonable to attribute 80% of the temperature change to the adiabatic lapse rate. Rephrase this sentence. I suspect it is a hold-over from the first version.

We thank the reviewer for catching and correctly identifying that. We removed the sentence.

Line 543: This approach absolutely needs to be dropped. If there is a potential range of temperature lapse rates, then you need to calculate the potential range of adiabatic temperature decreases based on that range in lapse rates. It is invalid to only use one value (the mean?) and then conclude that some small fraction of the total observed change must be due to non-adiabatic climate change. The best that could be argued is that the misfit _might_ be due to non-adiabatic temperature changes. But it might not be…

We agree. Confidently quantifying adiabatic and non-adiabatic contributions would require a more sophisticated approach than the "back of the envelope" calculation we had conducted. We have

therefore discontinued our approach and acknowledge that all the localized simulated temperature changes can be attributed to the adiabatic lapse rate (see our response to the comment in Line 552).

Line 552: No. The results suggest nothing of the sort. They suggest that all of the change can be attributed to adiabatic temperature changes, but that a small fraction _might_ be attributable to non-adiabatic changes. The signal is within the range of the noise.

We agree. We have modified the text to emphasize that all the localized temperature changes are due to the adiabatic lapse rate (see lines 550-561). We attribute only the adjacent far-field temperature changes to non-adiabatic-related processes, such as changes in the atmospheric circulation process.

*"The topography sensitivity experiments show significant localized changes in near-surface temperature. For all topographic configurations, the maximum changes were estimated in regions with modified topography, while changes in regions farther from the orogen are less pronounced. The less pronounced regional changes farther from the modified topography areas might be due to associated large-scale atmospheric changes and, therefore, caused by a non-adiabatic mechanism. However, these small and insignificant temperature differences may simply be due to modeling artifacts. On the other hand, the significant changes in regions of modified topography can mainly be attributed to the adiabatic temperature lapse rate, which defines how temperature changes with altitude. Although previous studies have indicated the possibility of non-adiabatic mechanisms (e.g., changes in tropospheric dynamics, local atmospheric humidity, and atmospheric circulation patterns) contributing to changes in addition to the adiabatic lapse rate changes (Ehlers and Poulsen, 2009; Feng and Poulsen, 2016; Kattel et al., 2015), an in-depth quantification of the relative contributions would be required to confidently attribute the changes to non-adiabatic processes."*

Line 677: I wonder if it would be worth highlighting the fact that the only way to get d18O values more negative than ~-8 per mil is to have topography that is higher than modern.

We agree with the reviewer and added a sentence with this statement in line 690.

*"We highlight that a magnitude of $\Delta\delta^{18}O_p$ value of -8 ‰, which is significant enough to be preserved in geologic archives, would only be achieved when the mean topography is higher than the modern Alps."*

Line 685: I don't see the 1 per mil per km. 0.5 per mil per km might be possible based on average values, but, again, the uncertainties are important.

The 1 ‰ and 0.5 ‰ in the Reviewer's comment can be derived from subtracting the modeled lapse rates in Figures 7A and 7B. We refer the reviewer to Fig. 7A for the 0.46 ‰/km difference between CTL and W2E1 for the western transect (i.e., -2.78 minus -2.32 ‰/km) and Fig. 7B for the 1.19 ‰/km difference between CTL and W2E1 for the northern transect (i.e., -3.37 minus -2.18 ‰/km). To avoid confusion for the reader, we have modified the text (see lines 695-699) to include the exact difference and the range of statistical uncertainties for the specific cases we highlighted in this section:

*"For instance, the W2E1 topographic configuration, which best matches the paleoelevation reconstruction in the Middle Miocene by Krsnik et al., 2021 would correspond to an increase of 0.46 (±0.15-0.24) and 1.19 (±0.09-0.11) ‰/km across the western and northern flanks compared to present-day topography. However, the estimated difference between W2E1 and the CTL across the southern flank is 0.2 (±0.13-0.16) ‰/km. This indicates that the impact on the isotopic lapse rate changes depends on the topographic rise and configuration, and the transect considered."*

Line 692: Without specifying what the effect of the model shortcomings are it is virtually impossible for most readers to consider these limitations in any meaningful way. I am not sure what to advise here, because I assume the effects of the model limitations are unquantified (and perhaps unquantifiable until better models are built). Nevertheless, this section reads very much like a disclaimer.

We added the model limitations in section 5.6 (lines 703-711) to highlight a certain degree of (unquantifiable) uncertainty associated with our simulation, as presented in previous studies (e.g., Langebroek et al., 2011; Werner et al., 2011). Quantifying the exact implications of these individual model limitations would likely require years (or decades) of model development and testing, expansion of $\delta^{18}O_p$ observations, and numerous sensitivity experiments that are not feasible to address in this manuscript (or within a single project or workgroup). We believe it is important to mention the possible sources of uncertainty for future studies. However, we emphasize that estimating the difference between the same specific iso-GCM outputs (and not observations) does not amplify any systematic biases or misrepresentations of the physics of the system; therefore, the presented signals should be robust.

Line 720: The limitations of this section come back to the uncertainties, which in this case should certainly be the prediction intervals. The primary question is whether, given a new data point or data set, you could distinguish between the proposed models. I suspect that given the spread in the data used to calculate lapse rates, the answer is no. However, the authors need to demonstrate that that is not the case.

We disagree. The primary focus of the linear approximation slope estimate between elevation and $\delta^{18}O_p$ values ("isotopic lapse rate") in this manuscript is to determine their potential changes compared to the estimates from the unmodified topographic configuration experiment. We do not intend to use these lapse rate models from our topographic sensitivity experiments as realistic predictive models for estimating paleoelevation. Instead, our goal is to compare these linear relationship estimates among the different experiments. Moreover, these estimates are based on simulated $\delta^{18}O_p$ values within a defined transect with fixed data points. Therefore, their estimated slope and ranges can only be applied to these specific simulation data points. The question raised by the reviewer about whether a new point or dataset would make a difference in the regression estimates is not relevant here, as the estimates are only related to fixed data points. Therefore, the uncertainty of the mean response for the fixed data points within the transect is what is important in this context. We acknowledge that the prediction interval (upper and lower limits) is larger, but its interpretation is limited to that specific regression line and cannot be transferred for comparison with others due to the different distribution of the fixed data points in the transect for a specific topographic scenario. We have added the prediction interval around each of the regression lines,

but its interpretation has less significance in this study. We have also modified the text in the methods section (lines 249-261) to clarify these points. If our explanation does not seem satisfactory to the reviewer, we respectfully ask them to clarify the mathematical basis and the reasons for suggestions so that we can efficiently resolve any concerns.

*"The elevation-$\delta^{18}O_p$ relationships, referred to here as the isotopic lapse rates (ILRs), were estimated for different geographic areas around the Alps (Fig. 1 A) by performing Ordinary Least Squares (OLS) linear regressions on the grid point values within each region. We use the notation -1‰/km (instead of 1‰/km) to report a decrease of 1‰ for an elevation increase of 1km. We highlight that the aim of the analysis is to determine if the elevation-$\delta^{18}O_p$ relationship over a specific transect would change in response to the different topographic configurations. The estimated lapse rates are not intended to serve as a predictive model for calculating paleoelevation but as a comparison among the topographic configurations to highlight the need to consider the potential changes in lapse rate through space and time. The statistical uncertainties of the calculated lapse rate are determined using the 95% confidence interval around the calculated OLS slope using t-distribution with n-2 degrees of freedom where the standard deviation of the slope is the point estimate for n data points. Additionally, the coefficient of determination ($R^2$), a measure for the fraction of the variability of the $\delta^{18}O_p$ values that can be explained by the best-fitted OLS estimates, is also reported. We further show the 95% confidence and prediction interval around the regression fitted model to highlight the uncertainties around the individual topographic configuration if it was meant to be used to calculate the paleoelevation for a reconstructed $\delta^{18}O_p$ values. In such a case, however, it would not be appropriate to compare the error limits around the regression line for the different scenarios, since their estimates are based on samples from different distributions. We refer the reader to Montgomery and Runger (2010) for more details about the mathematical derivation of the reported metrics."*

[Figure]

*Figure 6: Summer isotopic lapse rates (ILRs) estimates for the W1 topography scenarios (i.e., W1E0 (red), W1E2 (green)), and CTL (black) experiments for the different transects around the Alps as shown in E (West: 44-47 °N, 1-8 °E, south: 43-47 °N, 8-15 °E, and north: 47-50 °N, 5-16 °E). The ILRs are estimated as the $\delta^{18}O_p$-elevation gradients using linear regression. The lapse rate uncertainties are determined using the 95% confidence interval around the calculated OLS slope using t-distribution with n-2 degrees of freedom where the standard deviation of the slope is*

*the point estimate, coefficient of determination ($r^2$) is the measure for the fraction of the variability of the $\delta^{18}O_p$ values that can be explained by the best-fitted OLS estimates and the 95% confidence and prediction interval around the regression fitted model to highlight the uncertainties around the individual topographic configuration if it was meant to be used to calculate the paleoelevation for a reconstructed $\delta^{18}O_p$ values.*

[Figure]

*Figure 7: Summer isotopic lapse rates (ILR) estimates for the W2 topography scenarios (i.e., W2E0 (purple), W2E1 (gold)), CTL (black), and W2E2 (blue) experiments for the different transects around the Alps as shown in Fig. 6 E ((West: 44-47 °N, 1-8 °E, south: 43-47 °N, 8-15 °E, and north: 47-50 °N, 5-16 °E). The ILRs are estimated as the $\delta^{18}O_p$-elevation gradients using linear regression. The lapse rate uncertainties are determined using the 95% confidence interval around the calculated OLS slope using t-distribution with n-2 degrees of freedom where the standard deviation of the slope is the point estimate, coefficient of determination ($r^2$) is the measure for the fraction of the variability of the $\delta^{18}O_p$ values that can be explained by the best-fitted OLS estimates and the 95% confidence and prediction interval around the regression fitted model to highlight the uncertainties around the individual topographic configuration if it was meant to be used to calculate the paleoelevation for a reconstructed $\delta^{18}O_p$ values.*

Line 753: As I indicated in my first review, the conclusions and abstract need more detail and caveats associated with the conclusions, given the fact that these are the sections that most people will read. Stating that the changes are "distinctly different" does not communicate any of the nuances of potentially overlapping lapse rates in d8O or temperature.

We agree. We have modified the abstract to include a more nuanced presentation and discussion of the results, as suggested by the reviewer. We also expanded the conclusion to comment on the changes in $\delta^{18}O_p$ values. We now also mention the changes in temperature, precipitation, isotopic lapse rate, atmospheric circulation, and moisture transport that resulted in spatial changes of the $\delta^{18}O_p$ in response to the elevation changes (see lines 764-798).

*"The European Alps are hypothesized to have experienced diachronous surface uplift in response to post-collitional process such as slab break-off. Understanding the geodynamic and geomorphic evolution of the Alps requires knowledge of its surface uplift history. This study employs a model-based sensitivity analysis to investigate the regional climatic and $\delta^{18}O_p$ values response to diachronous surface uplift across the Alps. Overall, our results let us accept the hypotheses that the diachronous surface uplift of the West-Central and Eastern Alps would result in distinct regional climates and meteoric $\delta^{18}O_p$ patterns that differ from (1) present-day conditions and (2) conditions produced when the whole Alps are uplifted. If this signal is not lost during the formation of geological proxy material like paleosol carbonates, these records can be used in a stable isotope paleoaltimetry approach to test the hypothesis of eastward propagation of surface uplift in the Alps. We summarize the results as follows:*

1. *The diachronous surface uplift across the Alps significantly decreases $\delta^{18}O_p$ values up to ~8 ‰ over the modified areas, mainly due to an increase in orographic precipitation and adiabatic temperature lapse rate. The topographic scenarios with higher elevations in the West-Central Alps produce a greater decrease in $\delta^{18}O_p$ values and an expansion of the affected geographical domain surrounding the Alps when compared to present-day topography. The different topographic scenarios resulted in a less significant change in $\delta^{18}O_p$ values of 1-2 ‰ over the adjacent low-elevation areas around the Alps.*
2. *The $\delta^{18}O_p$ values changes were predominantly driven by the significant increase in precipitation amount of up to ~125 mm/month in response to surface uplift due to orographic airlifting and changes in precipitation dynamics. The surface uplift scenarios with higher West-Central Alps topography resulted in significantly drier conditions (rainshadow) over Northern Europe and towards the eastern flanks.*
3. *Surface uplift resulted in a localized decrease in near-surface temperature that also contributed to the decrease of $\delta^{18}O_p$ values. The temperature changes were only significant over the modified topographic areas, where they can be explained primarily by adiabatic temperature lapse rates. Smaller changes of up to -2 °C over regions farther from the Alps may be attributed to non-adiabatic processes, such as changes in atmospheric circulation.*
4. *The changes in elevation-$\delta^{18}O_p$ relationship (i.e., isotopic lapse rate) among the different topographic scenarios depend on the transect around the Alps and the magnitude of elevation changes. Some changes were small and within the statistical uncertainties' range. The differences in isotopic lapse rates are in the ranges of -0.24 to -0.83 (with the highest uncertainty of ±0.24), -0.17 to -1.19 (±0.14), and -0.15 to -0.94 (±0.16) ‰/km for the western, northern and southern transect, respectively. The differences in these estimates might be attributed to a different redistribution of precipitation and changes in moisture transport distance and pathways along specific transects.*

*Note that this study only quantifies the topographic signal while keeping paleoenvironmental conditions constant. Further experiments are needed to investigate the synergistic effects of combined topographic and paleoenvironmental changes and move towards plausible reconstructions of Alps topography and paleoclimate of specific times in the past. Furthermore, the next logical step to close the gap between the predicted meteoric $\delta^{18}O$ response and isotopic ratios extracted from archives is to employ proxy system models to investigate the signal transformation that takes place between these steps. This would allow*

*for a more accurate back-transformation that can ultimately refine paleoelevation estimates for the Alps."*

REFERENCES

Botsyun, S., Ehlers, T. A., Mutz, S. G., Methner, K., Krsnik, E., and Mulch, A.: Opportunities and Challenges for Paleoaltimetry in "Small" Orogens: Insights From the European Alps, Geophysical Research Letters, 47, e2019GL086046, https://doi.org/10.1029/2019GL086046, 2020.

Ehlers, T. A. and Poulsen, C. J.: Influence of Andean uplift on climate and paleoaltimetry estimates, Earth and Planetary Science Letters, 281, 238–248, https://doi.org/10.1016/j.epsl.2009.02.026, 2009.

Feng, R. and Poulsen, C. J.: Refinement of Eocene lapse rates, fossil-leaf altimetry, and North American Cordilleran surface elevation estimates, Earth and Planetary Science Letters, 436, 130–141, https://doi.org/10.1016/j.epsl.2015.12.022, 2016.

Feng, R., Poulsen, C. J., Werner, M., Chamberlain, C. P., Mix, H. T., and Mulch, A.: Early Cenozoic evolution of topography, climate, and stable isotopes in precipitation in the North American Cordillera, American Journal of Science, 313, 613–648, https://doi.org/10.2475/07.2013.01, 2013.

Insel, N., Poulsen, C. J., Ehlers, T. A., and Sturm, C.: Response of meteoric δ18O to surface uplift — Implications for Cenozoic Andean Plateau growth, Earth and Planetary Science Letters, 317–318, 262–272, https://doi.org/10.1016/j.epsl.2011.11.039, 2012.

Kattel, D. B., Yao, T., Yang, W., Gao, Y., and Tian, L.: Comparison of temperature lapse rates from the northern to the southern slopes of the Himalayas, International Journal of Climatology, 35, 4431–4443, https://doi.org/10.1002/joc.4297, 2015.

Krsnik, E., Methner, K., Campani, M., Botsyun, S., Mutz, S. G., Ehlers, T. A., Kempf, O., Fiebig, J., Schlunegger, F., and Mulch, A.: Miocene high elevation in the Central Alps, Solid Earth, 12, 2615–2631, https://doi.org/10.5194/se-12-2615-2021, 2021.

Lee, J.-E. and Fung, I.: "Amount effect" of water isotopes and quantitative analysis of post-condensation processes, Hydrol. Process., 22, 1–8, https://doi.org/10.1002/hyp.6637, 2008.

Montgomery, D. C. and Runger, G. C.: Applied Statistics and Probability for Engineers, John Wiley & Sons, 791 pp., 2010.

Risi, C., Noone, D., Frankenberg, C., and Worden, J.: Role of continental recycling in intraseasonal variations of continental moisture as deduced from model simulations and water vapor isotopic measurements: Continental Recycling and Water Isotopes, Water Resour. Res., 49, 4136–4156, https://doi.org/10.1002/wrcr.20312, 2013.

Langebroek, P. M., Werner, M., and Lohmann, G.: Climate information imprinted in oxygen-isotopic composition of precipitation in Europe, Earth and Planetary Science Letters, 311, 144–154, https://doi.org/10.1016/j.epsl.2011.08.049, 2011.

Werner, M., Langebroek, P. M., Carlsen, T., Herold, M., and Lohmann, G.: Stable water isotopes in the ECHAM5 general circulation model: Toward high-resolution isotope modeling on a global scale, J. Geophys. Res., 116, D15109, https://doi.org/10.1029/2011JD015681, 2011.

---

## Author Response (AR3)

Regarding: response to the reviews of our research article "**The effects of diachronous surface uplift of the European Alps on regional climate and the isotopic composition of precipitation**" by Boateng et al.

Dear Prof. Gabriele Messori,

We would like to thank you for the time and effort in handling our manuscript, and we extend our gratitude to Reviewer 1 (J. E. Saylor) for their invaluable suggestions that have significantly improved the quality of our manuscript. We also appreciate their dedication in reviewing it for the third time. We have meticulously proofread the manuscript and have addressed all the minor comments. The most significant changes are as follows:

1. We have added a few sentences to further clarify why the estimated lapses are not meant to constrain past vertical changes but rather to compare them with different configurations. This comparison highlights the non-stationarity of the lapse rate through time and space and violates the assumption of paleoaltimetry. This is because our sensitivity experiments solely address changes related to topography without considering corresponding global paleoenvironment changes such as paleogeography, atmospheric $CO_2$ levels, etc. We intend to address these additional changes in future studies, as emphasized in the manuscript.

2. We have also taken additional steps to clearly highlight the sections where we report the $\Delta\delta^{18}O_p$ (i.e., changes between the low- and high-elevation regions) and the differences in $\delta^{18}O_p$ values between the CTL and modified topographic experiments.

We have provided detailed information regarding the manuscript's revision in our point-by-point response to the Reviewer's comments. We deeply appreciate your efforts, as well as the reviewer, in helping us enhance our manuscript.

The submission file comprises our point-by-point response to the Reviewer's comments and the revised manuscript (with tracked changes) specifying all the modifications made in accordance with the minor comments. Please do not hesitate to contact us if further clarifications are required.

Sincerely,
Daniel Boateng (corresponding author), on behalf of all co-authors

Reviewer's comments are repeated in black. Authors' replies are highlighted in blue font, and the revised texts in the manuscript are in quotation marks with blue italics font.

**Review by J.E. Saylor**
**Summary**
This is my third review of the manuscript by Boateng et al. The authors have addressed all of the comments raised in the previous reviews. I understand the authors' motivations for

reporting confidence limits rather than prediction limits for the calculated isotopic lapse rates and the justification makes sense. Nevertheless, I appreciate that they include the confidence limits and thereby make this research as transparent as possible and also as useful as possible to future researchers.

**Recommendation**

I recommend that this manuscript be published after the authors address the minor comments below.

**General comments**

My comments below indicate that there are a disturbing number of minor issues with the manuscript considering that it is on its third round of reviews. I recommend that the authors carefully revise the text, critically looking for minor problems and internal inconsistencies. We appreciate the reviewer for bringing up these minor issues, and we have diligently addressed them.

**Detailed comments**

Line 18: Consider replacing "for stable" with "to be detected using stable".
Lines 109 & 111: No caps for "middle" in "middle Miocene" since it is not a formal epoch or age. Do a universal search and correct throughout the manuscript.
We have corrected the above comments.

Line 135–139: This is tricky because the atmospheric circulation interacts with topography as shown by this study. I am not sure what to recommend except to explicitly acknowledge the feedbacks between atmospheric circulation and topographic change.
We have extended the sentence to indicate that atmospheric circulation can change due to global and regional paleoenvironmental factors, including topography (see line 139).

Line 178: Delete, "The reader is advised that". It is condescending.
This has been corrected.

Line 183: What is the ECHAM5-wise model resolution?
We have added the model resolution (see line 183).

Line 252: Establishing lapse rates for paleoelevation studies may not be the goal of this study, it is an obvious application of this research. Would such an application be valid? If not, why not? This doesn't affect the current study, but an explanatory statement may help future researchers use your research appropriately.
We have added a sentence to clarify why estimates may be less suitable for lapse rate estimations, given that the experiment exclusively considers changes in topography without accounting for their corresponding global paleoenvironmental changes (see lines 250-253).

Line 259: I am confused by this statement. Are the authors stating that the lapse rates and associated uncertainties should not be used in an empirical paleoelevation study? If so, the

reasoning is not clear but is quite important to how this study is used in future research. Can you expand on the reasoning for this statement or clarify this statement?

We have extended the sentence to encompass the assumed conditions under which the lapse rate can be employed for paleoelevation calculations in future studies (see line 258).

Line 294: Is the offset systematic or random? I have never found the overlays such as those presented in Figure 1B a particularly helpful way to visually evaluate the data. I would recommend a more simple plot such as a biplot of GNIP d18O vs model d18O for each of the GNIP locations. It seems like that would be much easier to evaluate the deviation of the model from empirical data.

These deviations are systematic and have also been highlighted in previous studies (e.g., Langebroek et al., 2011; Werner et al., 2011). The reason for this comparison is to check how realistically the model simulates the spatial patterns of $\delta^{18}O_p$ and, therefore, evaluating the GNIPs with the regional patterns of the ECHAM5-wiso predictions help show the consistency of the model representing the spatial variability. Therefore, we would like to keep this type of plot in this manuscript.

Line 385: I see ~-6 per mil change between the CTL and W2E1 in Figure 5C at ~8 degrees E, but I am not sure that I see -8 per mill change anywhere. Can you clarify where this -8 per mil comes from?

We thank the reviewer for pointing this out. In this sentence, we are referring to the changes between the low- and high-elevation regions. We have revised the sentence to specify the $\Delta\delta^{18}O_p$ of up to -8 per mil, as illustrated in Fig. 5.

Line 387: But these values are slightly lower in Figure 5B (~-11, ~-9, and -6 per mil). How should the reader understand the difference between text and the figures?

We initially reported the values for the longitude transect. We have now updated the text to include the latitude transect and, as a result, correctly report the suggested changes.

Line 678: Doesn't this contradict the argument made in lines 135–139?

Line 678 pertains to the modified topography conditions, while lines 135-139 are related to the atmospheric control of present-day European climate.

Line 690: Shouldn't this be a d18O value (rather than Dd18O value) based on Figures 6 and 7?

We refer to $\Delta\delta^{18}O_p$ values here.

Line 773: Should this be Dd18O or d18O (see comment above on line 690)?

We refer to the $\delta^{18}O_p$ values difference here.